# Flood patterns in a catchment with mixed bedrock geology and a hilly landscape: identification of flashy runoff contributions during storm events

Audrey Douinot[1], Jean François Iffly[1], Cyrille Tailliez[1], Claude Meisch[2], Laurent Pfister[1]

[1]Environmental Research and Innovation Department (ERIN), Luxembourg Institute of Science and Technology (LIST), Belvaux, Luxembourg
[2]Administration de la Gestion de l'Eau - Division de l'Hydrologie, 1, avenue du Rock'n'roll, 4361 Esch-sur-Alzette, Luxembourg

*Correspondence to*: A. Douinot (audreydouinot@gmail.com)

**Abstract.**

With flash flood events having been repeatedly observed in Central and Western Europe in recent years, there is a growing interest in how catchment physiographic properties and hydrological conditions are eventually controlling rapid and concentrated hydrological responses. Here, we focus on a set of two nested catchments in Luxembourg (Europe) that have been exposed in 2016 and 2018 to flash flood events and study their seasonal runoff time transfer distributions. Both catchments are of similar size (~30 km$^2$) and have analogous hydrological distance distributions, but their geological bedrock and landscape features are notably different. The upper catchment (KOE) is dominated by a low land area (38% of the catchment are located less than 30 m above the river network) consisting of variegated marly bedrock (Midle Keuper Km3) and moderately steep Luxembourg sandstone outcrops (Lower Liassic Li2). The lower catchment (HM) has its drainage network deeply cut into the Luxembourg sandstone, with half of it being covered by marly plateaus (Lower Liassic Li3, located between 80 m and 100 m above the river network) featuring heavy clay soil. Based on data generated from a dedicated hydro-meteorological monitoring network, we calculated for 40 rainfall-runoff events observed between August 2019 and July 2021 the corresponding net rainfall transfer time distributions (TTDs) from the hillslopes to the catchment outlet. We then compared the TTD properties and related them to the catchment's hydrological state and rainfall properties. We observed a marked seasonality in TTDs for both catchments. The KOE catchment reacts fastest during the winter period (December - February), while its response time is most delayed and spread out during periods of catchment recharging (October – November) and drying (March - May). The HM catchment exhibits similar TTDs during the mid-October to mid-April period, but they diverge markedly duringthe remaining part of the year, with opposite variations. During the mid-April to mid-October period, the average response time increases progressively in the KOE catchment. This behaviour is in stark contrast to the HM catchment, where response times are significantly shorter (peak discharge delay time decreases by -70% ± 28%) and more concentrated (runoff volume occurring in one hour increases by +48% ± 87%) during the mid-April to mid-October, in comparison to the extended winter period. This opposite seasonality leads us to consider different control

factors of the runoff transfer processes in relation with the topographic and geological layout of the catchment areas. In the KOE catchment, we found the TTD to be essentially driven by onset and cessation of hydrological connectivity on the flat marly terrain – the latter operating like a variable contributing area in terms of deep soil storage dynamics (except for one summer event). The HM section exhibits contrasted TTDs throughout the year, suggesting threshold dependent hydrological processes. More specifically, particularly quick runoff transfers seem to dominate under dry conditions (mid-April to mid-October). Correlation analyses compared to the literature on runoff generation on the one hand and our descriptive knowledge of the catchments on the other hand suggest multiple causes for the triggering of these rapid flows. The fractured marly plateaus, but also the hydrophobic forest litter forming during dry conditions on steep slopes, stand as our main hypotheses in this respect. Moreover, the absence of a riparian zone, preventing any dampening of (observed) abrupt and massive flows during extreme precipitation events, seems as well to be a key feature of the rapid runoff transfer.

For improving our understanding and forecasting capabilities in Luxembourg (and more broadly in the nearby regions of Germany, Belgium and France with similar physiographic and climate conditions), we recommend further studies focusing on catchments with fractured bedrock and limited riparian zones. Special attention may equally be given to the hypothesized responses of hydrophobic soil surfaces on steep hillslopes and marly soils to heavy precipitation events occurring after extended dry spells.

## 1 Introduction

### 1.1 Background

One key aspect of flood risk management consists in determining vulnerable areas exposed to hydrological hazard. When affecting built areas, flash floods can be particularly destructive due to: i) their short time of occurrence that leaves very limited or no time to the population for protecting their lives and properties (e.g., evacuation of people and goods, flood fencing); ii) a very rapid concentration of water volumes, leading to high – or even extreme – flood peaks.

Such sudden and devastating flood events are commonly referred to as "flash floods". The non-exhaustive emergency events database (EM-DAT, www.emdat.be consulted on 27.07.2020) has reported no less than 550 fatalities, 616.760 affected inhabitants and 17.6 billion US$ of damage related to flash floods in Europe over the past 20 years. They have been extensively studied precisely because of their high destructive potential for exposed populations and infrastructures. More than 170 publications with the keyword "flash flood" have been listed in Scopus every year since 2015.

So far, studies on flash floods in Europe mainly focused on the Mediterranean area (MA) (Pereira et al., 2017; Llasat et al., 2016; Marchi et al., 2010; Ducrocq et al., 2014; Diakakis et al., 2017; Saber et al., 2018; Gaume et al., 2016). These studies show that the rainfall properties – more specifically the maximum amount of precipitation accumulated in a few hours – are of paramount importance for flash flood generation. However, many of these studies also pointed out the discrepancies of flash flood responses between catchments with contrasting geological substrate – the latter appearing to control the general flood shape, even in those very specific cases of quick storm flow generation processes (Payrastre et al., 2012; Vannier et al.,

2013; Douinot et al., 2018). Likewise, catchment water storage prior to these extreme events is determining the magnitude of the hydrological response (Massari et al., 2020; Tramblay et al., 2010; Berghuijs et al., 2019).

Headwaters are most prone to be impacted by flash flood type hydrological events. Orographic rainfall forcing can lead to intense and prevailing precipitation on catchments located at higher altitude. Steep hillslopes are intuitively perceived as contributing to a rapid concentration of the surface and subsurface flow, eventually leading to a quick transfer of runoff at event scale. Moreover, mountainous catchments may exhibit a more fractured bedrock, as they are subject to higher structural constraints (Miller and Dunne, 1996; Molnar, 2004; Slim et al., 2015). The numerous faults and cracks support quick water transfer through the weathered bedrock and explain fast hydrological responses, even though the soil can be highly permeable (Braud et al., 2016; Braud, 2015).

In recent years, flash flood events have been reported for catchments located in Central Europe (Ruiz-Villanueva et al., 2012; Van Campenhout et al., 2015; Bronstert et al., 2018; Bryndal et al., 2015). For example, two flash floods have occurred in 2016 and 2018 in Luxembourg (Pfister et al., 2018 & 2020). While the runoff coefficients determined for these events remained rather moderate (12% - 25%, Pfister et al., 2020), their almost instantaneous and non-attenuated hydrological response was very unusual for this physiographic and climate setting.

While most flash flood related literature published to date refers to the Mediterranean area (MA), the processes underlying flash floods in North Central Europe remain poorly understood. This mainly relates to the fact that in these catchments (i) the climate forcing is not primarily controlled by topography (as opposed to MA), (ii) catchment storage filling states are very different between early summer (storage levels being still high when flash floods occur in Central European catchments) and autumn (storage levels being low when flash floods occurr in MA catchments), and (iii) the underlying bedrock geology is very different between Central European and MA catchments.

Within North Central Europe, Luxembourg stands as an ideal hydrological test bed, located mostly inside the Moselle River basin. The country embraces a wide range of nested (headwater & mesoscale) catchments with various bedrock types and contrasted physiographic settings – covering a relatively small area (~ 2600 km$^2$) exposed to a rather homogenous pluvio-oceanic climate. The rainfall-runoff transformation has been extensively characterized and shows strong geological controls (Fenicia et al., 2014; Wrede et al., 2015; Pfister et al., 2017).

For a set of 16 nested catchments in Luxembourg, Pfister et al. (2017) reported very contrasted hydrological functions of water collection, storage and release. By leveraging 9 years worth of hydro-meteorological and stream isotopic data, they were able to document that a catchment's resilience to variable meteorological conditions is largely controlled by bedrock geology. Less permeable bedrock will lead to smaller catchment storage capacity, larger seasonal variability in runoff coefficients, and smaller baseflow mean transit times.

Wrede et al. (2015) and Fenicia et al. (2014) confirmed the threshold (or seasonally contrasted) behaviour of impermeable catchments. Using either a modelling framework over long-term time series or geochemical tracing of two events, they concluded that non-linear models are more appropriate for simulating rainfall-runoff responses, and that the pre-event water

proportions differ between seasons. Note that the catchment with a higher bedrock permeability (composed of sandstone) is characterized by a more stable reservoir that is reasonably well simulated by a linear model.

## 1.2 Status Quo

To date, all investigations focusing on rainfall-runoff transformation processes in the Luxembourg context have been limited to small experimental watersheds ($< 5$ km$^2$) or dedicated to catchment storage and release functions. While these studies have substantially improved our understanding of physiographic controls on runoff generation, we still have poor knowledge of the processes triggering flash flood events.

In flash flood prevention related research, the interest is not only set on runoff volumes, but also on the high reactivity, magnitude and intensity of the related hydrological response. Here, we ask – in the context of a North Central European study area – what is influencing the specific flash flood event patterns, beyond the extreme rainfall properties? We leverage prior work in our nested catchment set-up and explore if, how and to what extent catchment physiographical properties and hydrological states may eventually control – by dampening or enhancing – (i) mean transfer time and (ii) magnitude of hydrological responses in case of extreme precipitation events.

## 1.3 Hypotheses

Based on the current state-of-the-art on flash flood type events in Central Europe and MA regions, as well as on our recent findings on bedrock geology controls on fundamental catchment functions, we hypothesize that:

- Catchment bedrock geology is influencing – equally to what has been found for mean summer and winter runoff coefficients – flood hydrograph characteristics proper to intense summer storm events, similar to those typically found in the MA;
- Initial catchment storage - as translated by groundwater levels and soil moisture - alongside vegetation growing state, are important factors, controlling both the response time and the damping effect of the catchment, eventually worsening or mitigating the devastating potential of a flash flood.

## 1.4 Methodology

For testing our hypotheses, we compare the runoff transfer time distributions of two nested catchments in the Ernz Blanche basin (Luxembourg) – an area that has recently experienced several flash flood events. These two catchments have almost equal surface area, similar elevation ranges and hydrological distances, while their bedrock geology and physiographic features are very different. This makes them suitable candidates for comparing transfer time distributions (TTDs). We rely on a unit hydrograph model for calculating a TTD irrespective of the rainfall distribution. The model is applied on 40 moderate rainfall events that have occurred over two hydrological years (August 2019 – July 2021). We indeed conjecture that the hydrological responsiveness of a given catchment is detectable independently of the magnitude (i.e., volume and/or

intensity) of the precipitation event. The same model is also applied on the 2016 and 2018 flash flood events, with the aim of having reference transfer times characteristic of flash floods.

## 2 Study area and hydrological events

### 2.1 The Ernz Blanche catchment

The elongated Ernz Blanche catchment (102 km$^2$, approximatively 22.5 km long, 4.5 km wide) is located in eastern Luxembourg (Western Central Europe). This mesoscale catchment is part of the eastern limit of the sedimentary Paris basin - also called the Gutland area - where layers of permeable sandstone alternate with less permeable marls (Wrede et al., 2015). The elevation ranges between 190 m and 420 m.

The local climate is dominated by westerly atmospheric circulation and temperate air masses from the Atlantic (Pfister, Humbert & Hoffmann, 2000). Seasonal differences in air temperature measured over the period 1971–2000 range from 3.8 °C in winter (from October to March) to 14.3 °C in summer (from April to September) (Pfister et al., 2017). Average annual precipitation in the catchment is 853 mm.yr$^{-1}$ over the studied period (1$^{st}$ August 2019 - 1$^{st}$ August 2021). The spatial distribution of precipitation follows the topography, with annual rainfall totals decreasing from 890 mm on the high elevated plateaus to 760 mm around the catchment outlet (Reisdorf, Figure 1).

The Ernz Blanche catchment has been exposed to several flash flood events in the past (1958, 2016 and 2018). This area is representative of most physiographic features found in Luxembourg. With a view to study flash flood mechanistics, we have installed a multi-parameter monitoring network in June 2019, geared towards the study of extreme rainfall-runoff responses. Six stream-gauges have been installed along the 27.5 km long Ernz Blanche River (Douinot et al., 2019) – crossing two contrasted physiographical settings with a view to TTD comparison. In addition, four rain-gauges and soil moisture sensors were dispatched across the catchment to measure precipitation and soil water content, respectively (Figure 1). Three of the six stream-gauges – located at Koedange, Heffingen and Medernach – cut the Ernz Blanche catchment in two distinct sections: the Koedange subcatchment (KOE) and the Heffingen-Medernach section (HM). The two sections cover almost equal areas and exhibit similar elevation range and slope (table 1), but with different geological substrates and physiographic features. The area extending upstream of the Koedange station is almost equally split between variegated marly terrain (middle Keuper, Km3) which roughly delimits a flat area, and the Luxembourg sandstone outcrops (Li2, table 1). The area extending between the Heffingen and Medernach stations (hereinafter referred to as the HM catchment) mainly consists of deeply cut Luxembourg sandstone, the river network forming narrow valleys. A marly layer (Li3) partially overlays the sandstone and designs two elevated plateaus located on both sides of the Ernz Blanche (green features in Figure 1). The land uses follow the geological delineation: the Luxembourg sandstone substrate is essentially covered by forest while the marl substrates (Li3, Km3) are used for agriculture purposes (see the land uses in Figure S2, on supplementary materials).

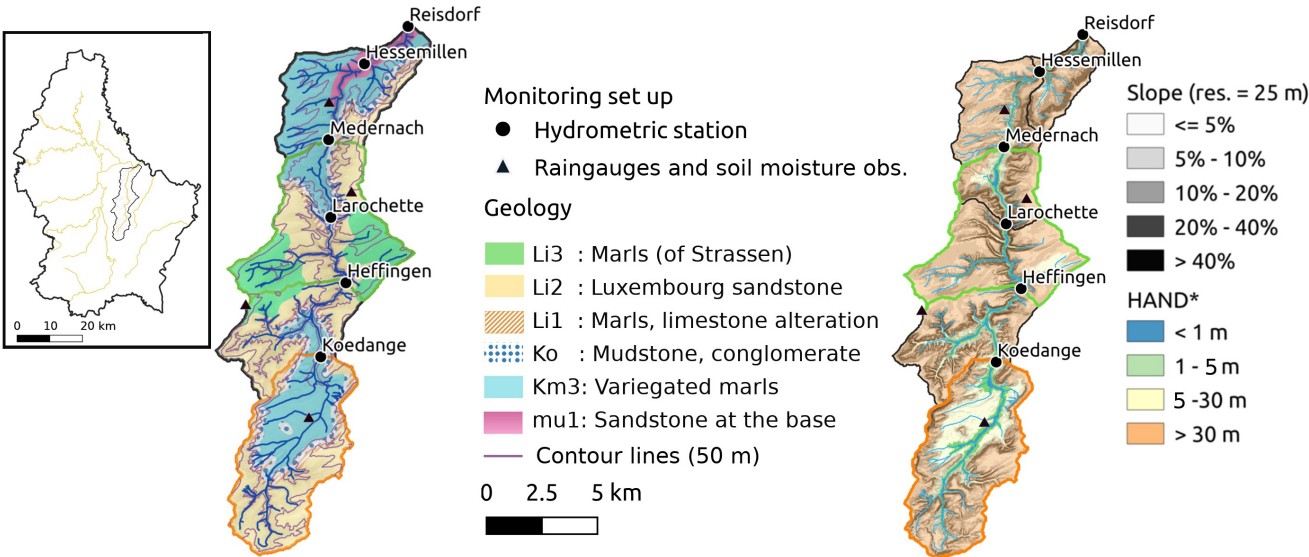

**Figure 1: Ernz Blanche catchment (102 km$^2$). Discharge and rainfall monitoring network; Left: geological characteristics (see Kausch & Maquil (2018) for more details). Right: topographical properties: slopes and Height Above the Nearest Drainage (HAND, Nobre et al., 2011). The Koedange subcatchment (KOE) and Heffingen-Medernach section (HM) are highlighted with orange and green contours respectively.**

**Table 1: Properties of the Ernz Blanche catchment by section**

| Catchment section | Area [km$^2$] | Elevation [m] $q^{25th}$ - $q^{75th}$ | Slope [%] $q^{25th}$ - $q^{75th}$ | Distance to the outlet [km] $q^{25th}$ - $q^{75th}$ | Area below 10 m height above the nearest drainage [km$^2$] | Lower geology | Outcropping/ Overlying geology |
|---|---|---|---|---|---|---|---|
| Koedange subcatchment | 31.14 | 320 - 385 | 3.55 – 12.5 | 3.91 – 8.57 | 4.6 (14.8 %) | Variegated marls Km3 (41.7%) | Lux. Sandstone Li2 (46.2%) |
| Heffingen - Medernach | 30.35 | 323 – 372 | 4.6 – 12.3 | 3.72 – 7.78 | 1.6 (5.3 %) | Lux. Sandstone Li2 (40.4%) | Marls of Strassen Li3 (35.2%) |

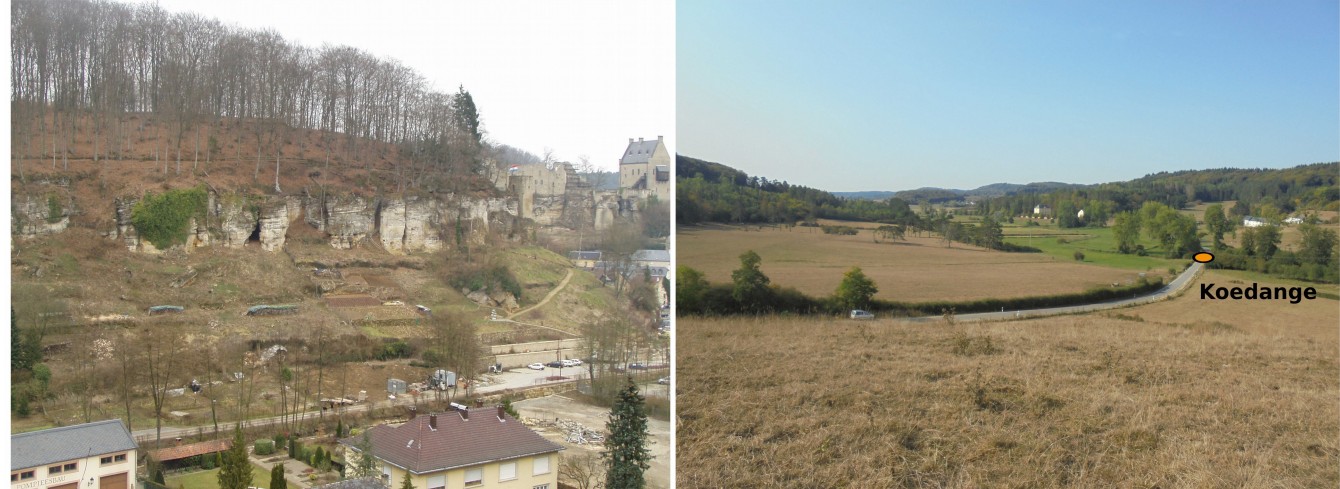

**Figure 2: Overview of the Ernz Blanche catchment. Left: View of the sandstone cliffs in the White Ernz valley at Larochette (Kausch & Maquil, 2018). Right: View the upstream part of the Koedange station (marked in orange). The arable land roughly corresponds to the Km3 geology, while the surrounding forest corresponds to the Li2 geology.**

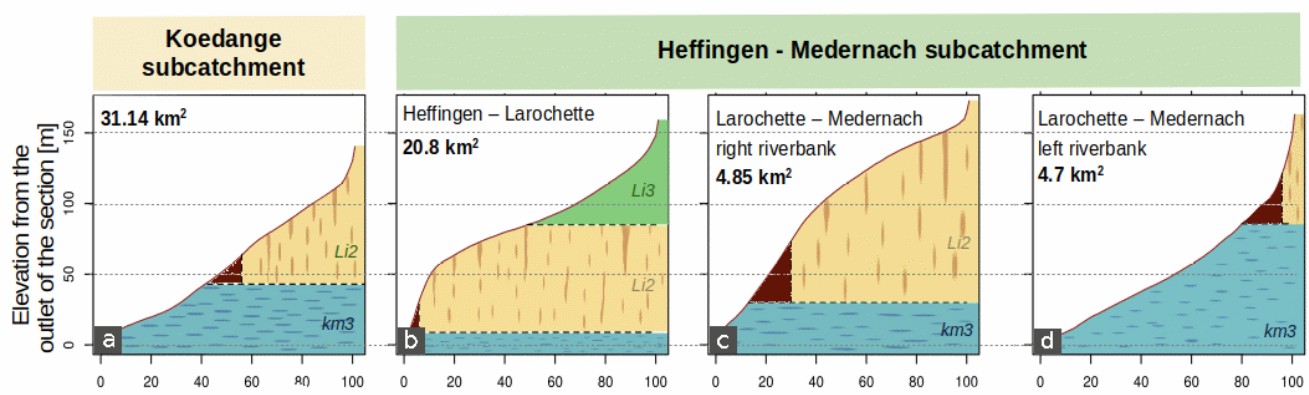

**Figure 3: Geological profiles in the Ernz Blanche catchment. Elevation distribution of a) the Koedange subcatchment, b,c,d) 3 subsections of the Heffingen-Medernach area: b) the Larochette – Heffingen subsection; c) the right riverbank of the Medernach – Larochette subsection; d) the left riverbank of the Medernach – Larochette subsection. Elevation is counted from the minimum elevation of each section. The geological substrates are designed according to their proportion in each section. Blue: Marls from middle Keuper (Km3); dark yellow: Luxembourg sandstone (Li2); green: Strassen marls (Li3), dark brown: conglomerates, marls and altered limestone.**

The similar elevation and slope characteristics actually hide contrasted landscape features (Figure 3). In the Koedange catchment (KOE, Figure 3-a) and on the left-handed hillslopes of the Ernz Blanche river between Medernach and Larochette (part of HM, Figure 3-d), the marly middle Keuper substrate is predominant and slopes are moderate (Figure 2, right). In the KOE catchment particularly, the marly middle Keuper substrate mainly forms a flat terrain in the vicinity of the river network and does not extend further than 30 m in height above the river network (table 1). On the Larochette-Heffingen section (Figure 3-b) and on the right riverbank of the Medernach-Larochette section (Figure 3-c) sandstone cliffs are more

prominent. The river network is deeply cut into the sandstone bedrock. Steep slopes close to the river network delineate
narrow valleys (Table 1 and Figure 1, right). As described in Kausch & Maquil (2018): "The Luxembourg Sandstone as a whole is cut through by a nearly vertical network of primary joints, with a meter- to decameter-wide spacing. These joints define large blocks or slabs and influence strongly the layout of the drainage system. Joints and fissures are mostly closed on the plateaus but may be widely opened by dissolution in lower lying zones of water infiltration or by unloading along the plateau edges [...]" (Figure 2, left).

### 2.2 Hydro-meteorological datasets (August 2019 - July 2020)

#### 2.2.1 The monitoring network

We leverage two years of rainfall and discharge measurements recorded at a 5-minute time step between 1st August 2019 and 1st August 2021. Rainfall has been recorded using 4 tipping bucket raingauges with an impulse of 0.2 mm (Campbell Kalyx, see figure 1 for the raingauge locations). The observed rainfall measurements were interpolated using the Thiessen polygon
method. The water levels have been recorded using a CS475A radar sensor. The discharge rating curves were determined via 20 gauging measurements per station, all carried out within the studied period. Note that the gauging campaigns also cover the two highest floods observed.

Soil humidity sensors (Campbell CS650) were installed at 20 cm and 50 cm depth next to the raingauge locations. They recorded soil humidity at a 5-minute time step in the 2 main soil textures of the catchment, namely sandy soils and clay soils.
The observed soil humidity measurements were weighted according to the cover rate of each soil texture to account for their spatial variability.

#### 2.2.2 Selection of the rainfall-runoff events and their characteristics

We selected 40 rainfall-runoff events (figure 4, table 2) according to the following criteria: i) the average rainfall amount based on data from the four  raingauges had to exceed 10 mm, and ii) there had to be less than 6 hours without rain within a
single event. The data set covers a wide range of rainfall event durations (table 2), spanning from several summer storms having lasted a few hours (with a minimum of 2.8 hours) to winter events spread over several days (the maximum being 6 days).  Aggregated five  minutes rainfall  varies from significant (i.e., up to 21.7 mm in 1 hour) to low (< 1.3 mm in 1 hour) rates. The seasonal cycle of the soil wetness state is also well represented by our dataset, with initial soil moisture conditions spanning almost the full width of the annual distribution $[q1^{th} – q99^{th}]$. Due to the large range of the observed rainfall forcing
and initial catchment wetness states, our two-year dataset covers a large diversity in floods. The runoff coefficients vary from 1.2% to 38.1%. The observed flood peaks span two orders of magnitudes.

Within our dataset, the extraordinary summer event of 13/07/2021 – which had dramatic consequences in the Greater Region (South Belgium, Eastern Germany) – consists of an extreme event in terms of rainfall amount (129 mm in 62 hours) and discharge peak (the highest water level was recorded during that event, ever since the installation of the oldest hydrometric

station at Larochette in 2014). Although runoff volumes are rather uncertain (table 1), the flood timing required for the methodology was recorded well enough to apply the unit hydrograph model. In addition, we selected the 2016 and 2018 flash flood events for which valuable discharge had been recorded at Larochette (69.4 km$^2$). We determined five minutes rainfall amounts from radar observations and raingauge measurement-based corrections. For both events, precipitation and the resulting floods relate to the catchment downstream of Heffingen (see the spatial rainfall amount patterns in Figure S1 on

supplementary materials).

Based on the rainfall properties and catchment states, the data set can be split in two categories related to the season of occurrence: the winter events occurring from October to April (i.e., when soil moisture reaches field capacity) are characterised by longer durations (Figure 4, left), while in summer (May – September), the rainfall intensities are higher. Among the observed hydrological responses (Figure 4, right) two moderate winter rainfall-runoff events (03/02/2020 &

21/12/2020) stand out with high discharge peaks, as well as the 2016 and 2018 summer events. The extraordinary event of 13/07/2021 is out of the frame of the PCA analysis, due to the related extreme rainfall amount and peak discharge. Note that the rainfall properties of the 2016 and 2018 flash floods do not appear that exceptional when compared to the data set of moderate events used in this study.

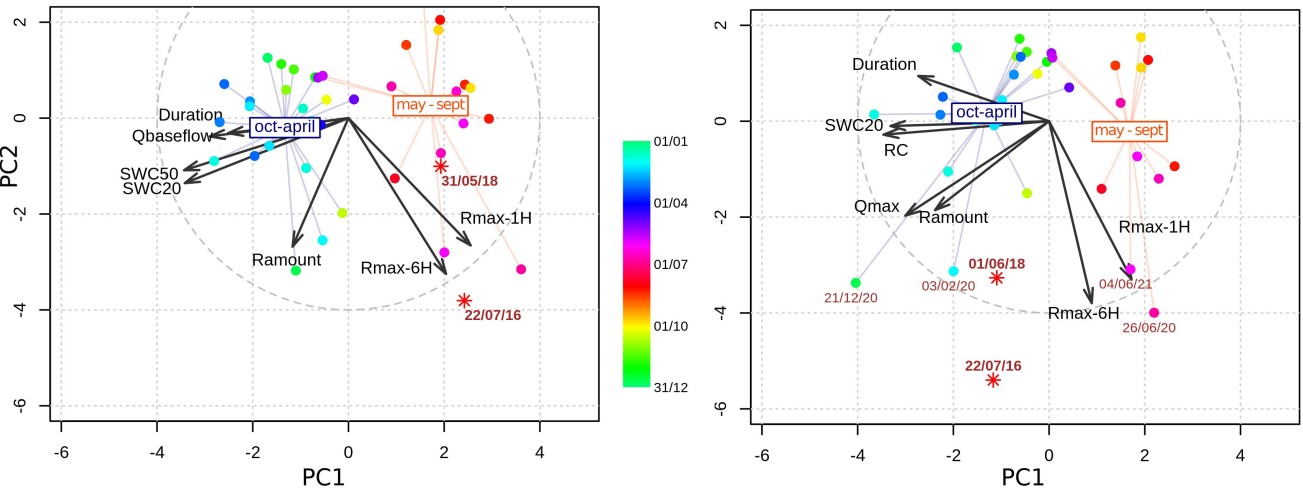

**Figure 4: Overview of the events from August 2019 to July 2021. Left: principal component analysis (PCA) taking into account rainfall properties and the wetness state of the Ernz Blanche catchment at Medernach. Right: PCA including hydrological response properties (Qmax: peak discharge and RC: Runoff Coefficient). The two flash flood events of 2016 and 2018 are positioned on the figure using the Larochette catchment data. The extrordinary event of the 13/07/2021 plots out of the lower left corner of the frame.**

From the discharge response visualizations (Figure 5), we were already able to discern two distinct patterns. The headwaters (as expressed through the Koedange and Heffingen stream gauges) consistently triggered rather attenuated hydrological responses. Further downstream, the stream gauges located downstream of Larochette exhibited a much more responsive behavioural pattern. The difference is most noticeable during summer.

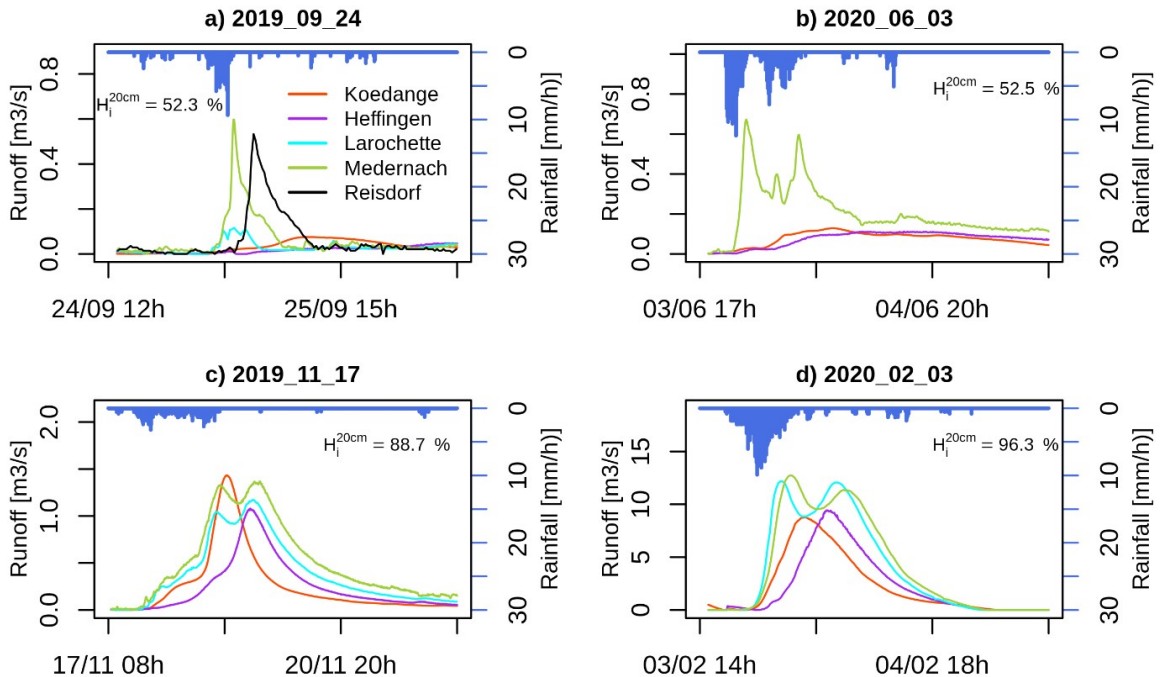

**Figure 5: Four rainfall-runoff events that have occurred in the Ernz Blanche catchment with different soil moisture conditions. Rainfall amounts are calculated for the Ernz Blanche catchment at Medernach (79 km²). The runoff time series are observed at Koedange (31.1 km² orange), Heffingen (48.8 km², purple), Larochette (69.4 km², cyan), Medernach (79 km², green) and Reisdorf (100.6 km², black). $H_i^{20cm}$ corresponds to the soil moisture conditions observed at 20 cm in depth before each event. Note that scales for discharge time series differently on each panel.**

**Table 2: Rainfall event properties, initial soil moisture and discharge characteristics. (\*) Rainfall statistics relate to the Medernach upper catchment. (\*\*) Initial soil moisture values correspond to the arithmetic mean of the four observed TS. (\*\*\*) RC: Runoff coefficient calculated for the Medernach upper catchment; Peak discharge: arithmetic mean of peak discharge observed at Koedange and Medernach. In bold: extreme values. (1) The peak discharge and the runoff coefficient was roughly estimated for the 13-07-2021 event. (2) The 2016 and 2018 flash flood events properties were assessed for the Ernz Blanche catchment at Larochette.**

| Event | Rainfall * | | | Soil moisture** [%] | | Runoff*** | |
|---|---|---|---|---|---|---|---|
| | Amount [mm] | Duration [h] | Max. intensity [mm/h] | -20cm in depth | -50cm in depth | RC [%] | Peak disch. [l.km⁻².s⁻¹] |
| 2019/08/06 | 11.1 | 14.5 | 3.45 | **52.6** | 70.1 | 1.44 | **2.4** |
| 2019/08/09 | 14.2 | 11.8 | 12.08 | 60.3 | 71.6 | 1.38 | 5.4 |
| 2019/08/12 | 13.5 | 7.9 | 8.21 | 54.4 | 72.8 | 1.65 | 5.7 |
| 2019/08/17 | 16.3 | 34.0 | 2.94 | 60.2 | 73.1 | 1.67 | 3.9 |
| 2019/09/24 | 11.0 | 27.9 | 4.12 | 52.5 | 70.6 | **1.24** | 5.0 |
| 2019/09/26 | **9.8** | 17.7 | 2.87 | 54.0 | 72.0 | 1.33 | 2.5 |
| 2019/10/07 | 30.3 | 67.1 | 3.51 | 84.2 | 77.9 | 2.30 | 6.7 |
| 2019/10/19 | 43.6 | 24.6 | 6.30 | 89.6 | 87.8 | 8.34 | 45.9 |

| | | | | | | | |
|---|---|---|---|---|---|---|---|
| 2019/11/02 | 21.4 | 74.4 | 3.35 | 91.0 | 88.7 | 10.00 | 19.1 |
| 2019/11/17 | 17.9 | 37.3 | 2.01 | 88.1 | 89.9 | 16.69 | 31.6 |
| 2019/11/26 | 17.3 | 60.5 | 1.99 | 90.6 | 89.2 | 14.12 | 22.5 |
| 2020/01/26 | 35.3 | 49.7 | 6.05 | 88.1 | 89.1 | 25.09 | 126.2 |
| 2020/01/31 | 21.5 | 35.7 | 6.56 | 88.0 | 89.8 | 23.18 | 98.1 |
| 2020/02/03 | 30.6 | 29.0 | 8.38 | 96.0 | 91.7 | 24.25 | 221.5 |
| 2020/02/09 | 26.2 | 53.6 | 4.57 | 89.2 | 91.7 | 19.76 | 77.9 |
| 2020/02/29 | 20.5 | 39.0 | 3.17 | **96.3** | 92.0 | 12.82 | 52.9 |
| 2020/03/04 | 24.2 | 45.4 | 2.42 | 93.3 | 93.8 | 30.34 | 126.7 |
| 2020/03/09 | 11.6 | 41.2 | 2.16 | 92.6 | 93.0 | 19.52 | 33.7 |
| 2020/04/29 | 35.2 | 69.8 | 4.48 | 57.9 | 77.6 | 2.12 | 5.4 |
| 2020/06/03 | 27.3 | 19.6 | 7.88 | 52.8 | 69.6 | 1.42 | 6.3 |
| 2020/06/12 | 16.6 | 16.0 | 9.74 | 60.5 | 70.1 | 2.65 | 8.8 |
| 2020/06/17 | 17.2 | 13.3 | 6.21 | 67.4 | 72.4 | 2.25 | 9.2 |
| 2020/06/26 | 28.4 | 20.3 | **21.68** | 63.8 | 71.9 | 3.99 | 27.7 |
| 2020/09/26 | 22.3 | 13.8 | 4.70 | **47.7** | **65.5** | 1.46 | 8.3 |
| 2020/12/02 | 17.4 | 43.3 | 3.13 | 89.3 | 84.3 | 8.47 | 32.0 |
| 2020/12/21 | 57.4 | 85.7 | 7.08 | 88.9 | 87.6 | 29.07 | **227.3** |
| 2020/12/27 | 14.0 | 56.3 | **1.26** | 89.6 | 92.5 | **38.08** | 77.29 |
| 2021/01/21 | 20.9 | 18.0 | 3.75 | 93.0 | 88.9 | 28.77 | 112.7 |
| 2021/01/27 | 43.5 | 129.7 | 3.49 | 90.0 | 88.2 | 34.80 | 87.0 |
| 2021/02/02 | 21.2 | 42.3 | 4.80 | 94.3 | **95.2** | 24.21 | 113.0 |
| 2021/02/06 | 13.0 | 17.4 | 4.55 | 92.6 | 94.7 | 31.55 | 73.7 |
| 2021/03/11 | 38.5 | **154.2** | 4.08 | 84.0 | 84.9 | 15.02 | 37.6 |
| 2021/04/09 | 36.8 | 46.7 | 3.35 | 79.1 | 83.2 | 13.18 | 52.5 |
| 2021/05/14 | 26.0 | 86.7 | 3.25 | 69.7 | 80.9 | 2.42 | 4.4 |
| 2021/05/24 | 25.9 | 53.6 | 3.10 | 78.0 | 80.6 | 4.50 | 10.0 |
| 2021/06/04 | 23.4 | **2.8** | 15.39 | 81.0 | 84.0 | 6.70 | 36.6 |
| 2021/06/19 | 15.8 | 6.7 | 11.53 | 63.1 | 83.7 | 1.96 | 7.4 |
| 2021/06/24 | 17.0 | 12.3 | 5.27 | 65.4 | 83.5 | 2.60 | 9.8 |
| 2021/07/13 | **128.9** | 62.1 | 15.49 | 87.0 | 86.3 | 22-30[1] | 400-600[1] |
| 2021/07/27 | 20.6 | 7.7 | 10.65 | 81.0 | 86.6 | 9.02 | 50.5 |
| **EXTREMA** | **9.8 – 128.9** | **2.8 – 154.2** | **1.26 – 21.68** | **47.7 – 96.3** | **65.5 – 95.2** | **1.24 – 38.08** | **2.4 – 227.3** |
| **2016/07/22**[2] | 23.9 | 12 | 5.03 | NA | NA | 12 - 16 | 210 - 260 |
| **2018/06/01**[2] | 43.0 | 22 | 12.2 | NA | NA | 19.0 - 22.0 | 170 - 200 |

## 3. Methodology – the unit hydrograph model

### 3.1 Modeling the rainfall-runoff transformation with a Gamma distribution function

We applied a simple unit hydrograph model to reproduce the hydrological responses of each rainfall forcing over each
catchment section. The unit hydrograph model assumes (by definition) that each net rainfall unit has the same TTD. We
assume that the runoff coefficient (RC) is constant during the event, and we thus consider our catchment in steady state. This
strong assumption prevents us from imposing a transient phase (variable RC and TTD) that we cannot measure.

Applying a unit hydrograph model allows for calculating a TTD independently of the rainfall distribution. Moreover, the
hydrological response of the HM section can be extracted from that determined for the entire Medernach catchment. We
chose the Gamma probability density function (PDF) as unit hydrograph model. The Gamma PDF enables a wide range of
likelihood TTD (Hrachowitz et al., 2010), while only requiring the calibration of two parameters.

The Heffingen-Merdernach catchment section requires an additive modelling unit to simulate the hydraulic transfer of the
discharge inflow from Heffingen. We chose a Gumbel PDF to simulate the 7.9 km hydraulic transfer from Heffingen to
Medernach. The hydraulic transfer process is indeed linear enough to be well simulated by this function. Two unit
hydrograph models and one hydraulic transfer model are applied to simulate the discharge at Koedange and Medernach
stations as described in equations 1 and 2.

$$Q(t)_{\text{Koedange}} = \int_0^t R_{\text{Koedange}}(\tau)\,\text{Ga}^R_{\mu,\theta}(t-\tau)\,\mathrm{d}\tau \tag{1}$$

$$Q(t)_{\text{Medernach}} = \int_0^t R_{\text{Heffingen}-\text{Medernach}}(\tau)\,\text{Ga}^R_{\mu,\theta}(t-\tau)\,\mathrm{d}\tau + \int_0^t Q_{\text{Heffingen}}(\tau)\,\text{Gu}^Q_{\mu,\theta}(t-\tau)\,d \tag{2}$$

With: $R_x(t)$ is the net rainfall amount after infiltration on the X (either KOE or HM) catchment section; $Q_{\text{Heffingen}}(t)$ is
the discharge observed at Heffingen station; $Ga^R_{\mu,\theta}(t)$ is the Gamma PDF modelling the transfer time distribution of
$R_x(t)$; $Gu^Q_{\mu,\theta}(t)$ is the Gumbel PDF modelling the hydraulic transfer of the catchment inflow at Heffingen. $(\mu,\theta)$ are the
model parameters.

The Gamma and the Gumbel PDF are described in equations 3 and 4, respectively:

$$\text{Ga}_{\mu,\theta}(t) = \frac{1}{\Gamma(\mu)}\,e^{\frac{-t}{\theta}}\cdot t^{\mu-1} \quad \text{where } \Gamma(\mu) \text{ is the gamma function} \tag{3}$$

$$\text{Gu}_{\mu,\theta}(t) = \frac{1}{\theta}\cdot\exp\left(\frac{-t-\mu}{\theta}+e^{\frac{-t-\mu}{\theta}}\right) \tag{4}$$

For each event, the net rainfall amount after infiltration - $R_x^{evt\_i}(t)$ – is assessed from the observed runoff coefficient ($RC_x^{evt\_i}$)
as described in equation 5,6,7.

$$RC_{MH}^{evt-i} = \frac{\int_{t_{init}}^{t_{end}} Q_{Medernach}(t) - Q_{Medernach}(t_{init})\,dt - \int_{t_{init}}^{t_{end}} Q_{Heffingen}(t) - Q_{Heffingen}(t_{init})\,dt}{\sum_{t_{init}}^{t_{end}} P_{Medernach-Heffingen}(t)} \qquad (5)$$

$$RC_{K}^{evt-i} = \frac{\int_{t_{init}}^{t_{end}} Q_{Koedange}(t) - Q_{Koedange}(t_{init})\,dt}{\sum_{t_{init}}^{t_{end}} P_{Koedange}(t)} \qquad (6)$$

$$R_x^{evt-i}(t) = RC_x^{evt-i} \cdot P_x(t) \qquad (7)$$

With: $P_x(t)$ and $R_x(t)$ is the rainfall amount and the net rainfall amount respectively observed in the $X$ (KOE or HM) catchment section; $t_{init}$ and $t_{end}$ the start and the end time of the event $evt$-$i$, and $RC_x^{evt,i}$ the observed runoff coefficient during the event $evt$-$i$ in the $X$ catchment section.

We relied on a Monte Carlo analysis with 2000 parameter sets for calibrating the models. The models' parameter ($\mu,\theta$) ranges are presented in table 3. They have been chosen according to prior rough assessments of the median transfer time (period between the median times of the net rainfall and the runoff distribution, see supplementary material S3) and the time lag between flood peaks at Medernach and Heffingen (for the hydraulic model).

**Table 3: Model's parameter ranges**

|  | μ | θ |
| --- | --- | --- |
| Koedange model (Gamma PDF) | 1 – 18 | 0.1 – 15 |
| Heffingen-Medernah model (Gamma PDF) | 0.1 – 16 | 0.1 – 15 |
| Hydraulic model (Gumbel PDF) | 0.1 – 4.5 | 0.1 – 5 |
| Larochette model (Gamma PDF) | 0.1 – 16 | 0.1 – 15 |

We applied the unit hydrograph model to the 2016 and 2018 flash flood events at Larochette, similar to modelling of the KOE catchment. Although the modelling covers the hydrological response of the entire 69,4 km$^2$ of the catchment at Larochette, we assume comparable transfer times – and therefore comparable parameter ranges – because of the precipitation during these two events being located in the first half of the catchment (see rainfall pattern in supplementary materials, Figure S1).

For our event-based calibration, we used the Root Mean Square Error (RMSE) as objective function. It enables to focus the calibration on the high flows and their timing (unlike an objective set on the flow duration curve for example). From the calibration results, we first select the 50 best simulations. We then gradually reduce the number of acceptable simulations, as the variation of the RMSE scores among this likelihood subset exceeds 10 % of the mean discharge. This limit ensures

homogeneous modelling results within the subset, so that they could consequently be equally considered. (Note that a
weighting process according to the RMSE could also have been chosen for similar results).

**3.2 Properties of the transfer time distributions and correlation analysis**

From the event-based calibration, we obtain a TTD set for each event over each catchment section. We opted for comparing the different TTD sets by defining three properties (Figure 6):

- TTD50: the median transfer time [h], i.e. the 50[th] percentile of the TTD;
- TTDpk: the flow peak lag time [h], i.e. the time where the TTD is at its maximum;
- VOL1h: the runoff response concentration in one hour [% of the total runoff volume].

TTD50 is representative of the time lag between the hyetograph and hyetogram barycenter, which characterizes the average transfer speed of a catchment. TTDpk and VOL1h characterize the dominant transfer speed and how the transferred water volume is more or less concentrated around the flood peak. The two latter properties are of first order of interest to
characterize the ability of a catchment to generate fast and high magnitude floods, and eventually flash floods.

We analyze the variation of the TTD properties according to the different rainfall and catchment properties. Among a larger number of rainfall properties, we chose: the rainfall amount (Rcumul [mm]), the rainfall duration (Rduration [h]), the maximum rainfall intensity in 1 hour (I1h [mm.h$^{-1}$]), the mean rainfall intensity (Imean [mm.h$^{-1}$]). Those statistics were picked from a larger number of options, appearing during the analysis to be the most significant. The catchment state before
each hydrological event is described using: the soil moisture at -20 cm depth (SWC20 [%]), and at -50 cm in depth (SWC50 [%]), the baseflow (Qbase [m$^3$.km$^{-2}$.s$^{-1}$]), and the calendar day (DAY). The latter is assessed from a Joint Research Center dataset (Pistocchi, 2015), which provides the 2002-2006 monthly average (1 km$^2$ resolution), and which has been linearly interpolated to get daily values. The different statistics were chosen because of their availability and as they enable to characterize catchment storage state (Qbase); soil moisture states (SWCx) and seasonal time (DAY) .

The dependency of the TTDs versus the rainfall and catchment state properties is studied through the non-parametric correlation scores Kendall's τ (Kendall, 1938) and Hoeffding's *D* (Hoeffding, 1948). Both are rank-based approaches. Kendall's τ assesses the possible monotonic relationship between two variables, including non-linear relations (unlike the Pearson coefficient). Hoeffding's D can detect non monotonic relationships. The statistics are calculated using Stats (3.4.4) and Hmisc (4.4-0) packages on R.

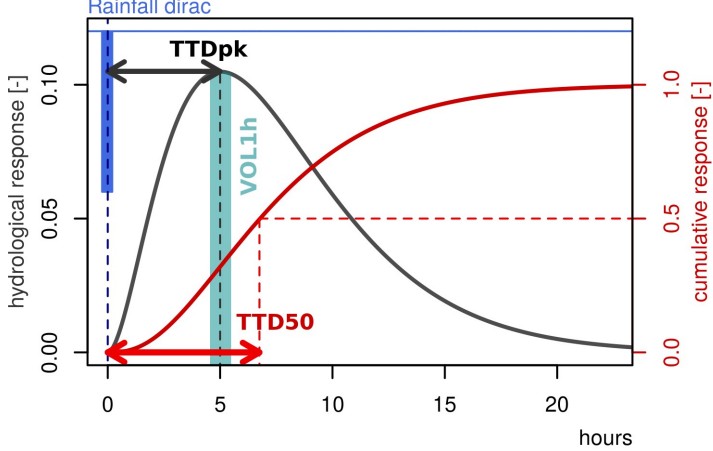

**Figure 6: Illustration of the TTD properties on a unit hydrograph: TTD50, TTDpk and VOL1h.**

## 4 Results

### 4.1 Validation of the models

Table 4 provides a multiple assessment of the model calibrations using the Root Mean Square Errors of the event times series (RMSE), as well as of the flow duration curve (FDC), and the Nash-Sutcliffe coefficient (NASH).

According to the Nash coefficient, the models fitted very well all events, except one (12-06-2020) on the Heffingen-Medernach section. Most of the RMSE scores are below 15% of the maximum peak discharge – which is an acceptable result – except for one event on the Heffingen-Medernach section (HM) and one event on the Koedange subcatchment (KOE). The latter corresponds to one of the smallest events in terms of flood peak which make it sensitive to this assessment. The simulation for the Heffingen-Medernach section was rather poor for a 3-peaked flood event that had occurred on 29[th] February 2020.

According to the flow duration curve assessment, the models show limitations for simulating three summer events with high rainfall intensity on HM, two large winter events on KOE occurring while water storage was high but not yet at maximum levels, and four summer events.

**Table 4: Assessment of the models' calibration. Median score of the likelihood selected simulations: RMSE = Root Mean Square Error expressed as a percentage of the observed peak discharge; NASH = Nash-Sutcliffe coefficient; FDC = Root Mean Square Error of the flow duration curve expressed as a percentage of the mean discharge. Bad scores are highlighted in bold.**

| Event | KOE | | | HM | | |
|---|---|---|---|---|---|---|
| | **RMSE [% maxQ]** | **NASH [ - ]** | **FDC [% meanQ]** | **RMSE [% maxQ]** | **NASH [ - ]** | **FDC [% meanQ]** |
| 2019/08/06 | 8.2 | 0.93 | 11.6 | NO DATA | | |
| 2019/08/09 | **16.6** | 0.75 | 24.6 | | | |
| 2019/08/12 | 11.9 | 0.87 | 19.2 | | | |

| | | | | | | |
|---|---|---|---|---|---|---|
| 2019/08/17 | 6.8 | 0.95 | 11.9 | | | |
| 2019/09/24 | 7.4 | 0.95 | 14.7 | 5.3 | 0.87 | **39.6** |
| 2019/09/26 | 8.1 | 0.94 | 10.5 | 10.5 | 0.80 | 19.0 |
| 2019/10/07 | 9.9 | 0.86 | 19.3 | 9.4 | 0.85 | 7.9 |
| 2019/10/19 | 8.8 | 0.94 | 18.1 | 7.8 | 0.93 | 10.0 |
| 2019/11/02 | 14.7 | 0.78 | 31.4 | 6.8 | 0.94 | 9.7 |
| 2019/11/17 | 10.5 | 0.81 | **50.9** | 9.3 | 0.90 | 16.4 |
| 2019/11/26 | 10.6 | 0.84 | 14.3 | 5.9 | 0.96 | 9.6 |
| 2020/01/26 | 11.2 | 0.83 | **38.9** | 8.3 | 0.91 | 14.4 |
| 2020/01/31 | 7.6 | 0.93 | 16.3 | 8.1 | 0.94 | 13.0 |
| 2020/02/03 | 4.8 | 0.98 | 15.4 | 10.4 | 0.91 | 21.7 |
| 2020/02/09 | 6.7 | 0.94 | 16.2 | 7.5 | 0.94 | 8.0 |
| 2020/02/29 | 10.0 | 0.89 | 25.9 | **17.8** | 0.63 | 26.3 |
| 2020/03/04 | 5.1 | 0.97 | 7.7 | 4.6 | 0.98 | 8.9 |
| 2020/03/09 | 9.2 | 0.88 | 27.5 | 9.1 | 0.90 | 10.0 |
| 2020/04/29 | 7.4 | 0.93 | 11.8 | 9.1 | 0.77 | 10.0 |
| 2020/06/03 | 12.5 | 0.79 | 14.4 | 9.9 | 0.74 | 23.2 |
| 2020/06/12 | 7.5 | 0.93 | 15.7 | 12.3 | **0.18** | **39.1** |
| 2020/06/17 | 8.8 | 0.92 | 18.0 | 9.7 | 0.80 | 19.7 |
| 2020/06/26 | 8.6 | 0.85 | **42.6** | 10.4 | 0.60 | **46.0** |
| 2020/09/26 | 4.9 | 0.98 | 8.4 | 5.5 | 0.93 | 10.9 |
| 2020/12/02 | 5.5 | 0.97 | 12.0 | 6.9 | 0.96 | 12.2 |
| 2020/12/21 | 6.8 | 0.92 | 21.1 | 6.2 | 0.92 | 25.0 |
| 2020/12/27 | 6.0 | 0.95 | 14.6 | 7.8 | 0.91 | 11.9 |
| 2021/01/21 | 8.0 | 0.94 | 14.3 | 10.5 | 0.93 | 11.6 |
| 2021/01/27 | 7.6 | 0.93 | 9.1 | 9.0 | 0.91 | 13.0 |
| 2021/02/02 | 8.8 | 0.91 | 18.0 | 9.0 | 0.94 | 19.1 |
| 2021/02/06 | 7.2 | 0.95 | 13.2 | 10.9 | 0.92 | 12.6 |
| 2021/03/11 | 11.6 | 0.69 | 26.8 | 10.1 | 0.86 | 17.5 |
| 2021/04/09 | 14.1 | 0.64 | **82.6** | 8.2 | 0.88 | 20.6 |
| 2021/05/14 | 11.7 | 0.80 | 18.9 | 10.9 | 0.74 | 10.3 |
| 2021/05/24 | 12.7 | 0.69 | **36.7** | 10.2 | 0.84 | 9.6 |
| 2021/06/04 | 8.4 | 0.91 | 33.8 | 8.1 | 0.90 | 15.9 |
| 2021/06/19 | 11.1 | 0.81 | 26.7 | 10.2 | 0.70 | 36.9 |
| 2021/06/24 | 8.1 | 0.91 | 16.1 | 7.5 | 0.85 | 29.3 |
| 2021/07/13 | 8.6 | 0.82 | **60.4** | 5.3 | 0.94 | **39.5** |
| 2021/07/27 | 9.7 | 0.93 | 15.3 | 8.1 | 0.93 | 8.6 |
| 2016/07/22 | | At Larochette : | | 12.0 | 0.77 | 14.1 |
| 2018/05/31 | | | | 14.6 | 0.76 | **35.9** |

Figure 7 shows three event simulations for the KOE catchment. The events were chosen as representative of the event set

simulations. The simulations of the 26/01/2020 event for the Koedange subcatchment (Fig. 7a) overestimate the rising limb and underestimate and delay the flood peak. This limitation of the model is indicated by the low FDC values. While the first part of the flood is overestimated, the second part with the major peak is slightly underestimated. The second batch of examples (Fig. 7b) shows well simulated events for KOE, where the flood pattern is well reproduced, despite the strong heterogeneity of the rainfall. The particular case of the 26/06/2020 event is shown in figure 7c. This event consisted in 2 consecutive storms, the first one having the highest intensities of the entire time series. Here the simulations "do a

compromise" for simulating both flood peak responses: the first one tends to be underestimated, while the second one is overestimated. We can also notice that only a few simulations have been validated.

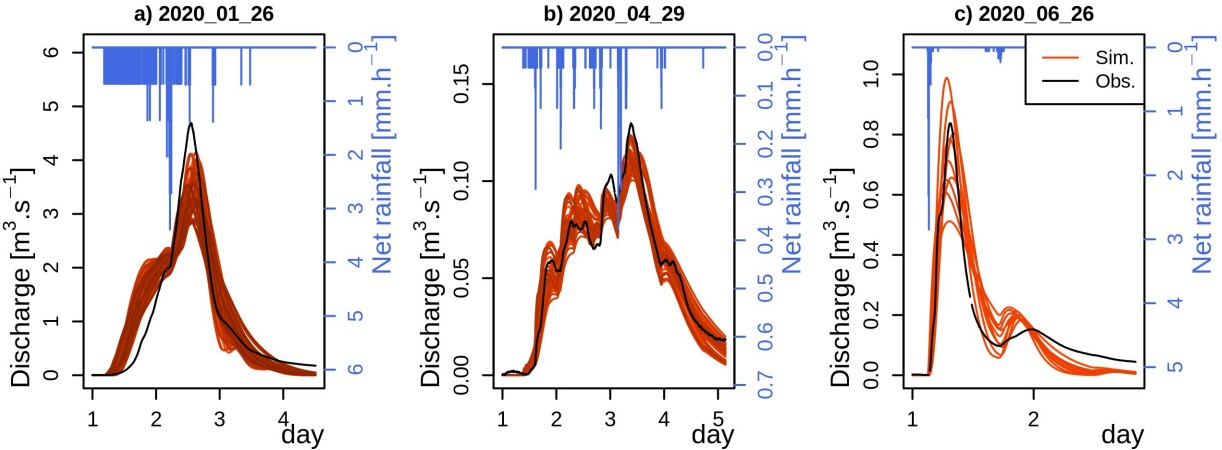

**Figure 7: Examples of simulated events for the Koedange subcatchment. Event a) is representative of the winter event simulations. The displayed event on panel b) is representative of the well simulated events. On panel c) is displayed the summer event on 26/06/2020, where the peak discharge tends to be underestimated while the second moderate event is overestimated.**

Figure 8 shows three event simulations for the HM section. Similar to the KOE catchment, the simulations tend to overestimate the rising limb and to underestimate the flood peak for autumn and early winter events - but to a smaller extent (Fig. 8a). The 29/04/2020 event displayed on panel b in Figure 8 is representative of the well simulated events for the HM section. It shows how well the overall flood pattern is simulated. Note that for the HM section the instantaneous flood peaks observed during the early stages of the rising limb, are not reproduced by the simulations. Those peaks last little more than

two or three 5-minute time steps, which explains why the scores are not affected by these model limitations (the errors calculated on a couple of time steps are dissolved within the overall TS assessment).

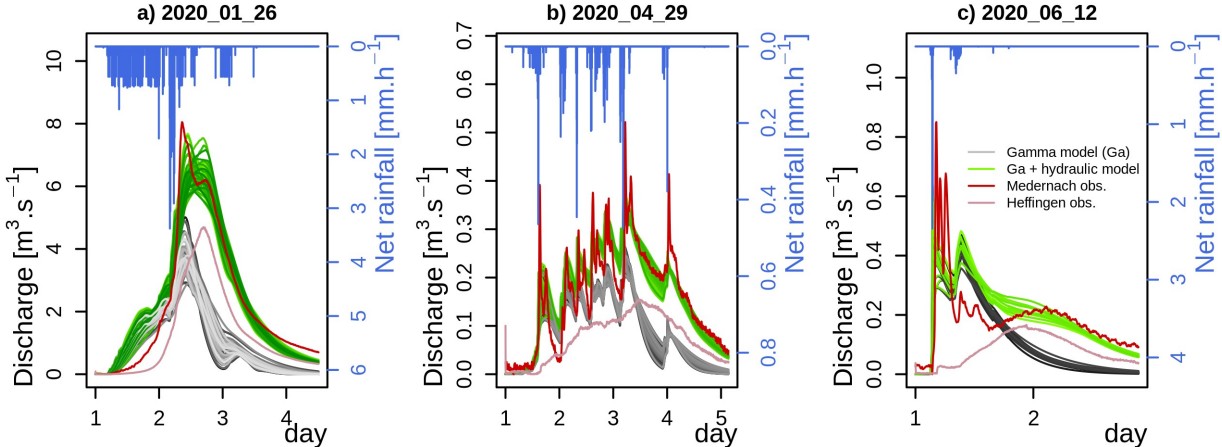

**Figure 8: Examples of simulated events on the Heffingen-Medernach catchment section. The grey lines correspond to the HM runoff transfer only, while the green lines correspond to this runoff transfer + the hydraulic transfer of the Heffingen inflow.**

The 12/06/2020 event displayed in Figure 8c), alongside three short storm events that occurred on 19/06/2021, 27/07/2021

and 04/06/2021, show the models' limitations. A three-peaked observed response is caused by a high intensity and short rainfall forcing. Note that there is reportedly no error in the one peak rainfall observation.

## 4.2 Comparison of Koedange and Heffingen-Medernach TTDs

We observed a large diversity in TTDs, as obtained after the event-based calibration for the HM section and KOE catchment (Figure 9; Table 5). The median transit times (TTD50) vary between 2.0h and 23.9h, the lag time between the rainfall unit

occurrence and the peak response (TTDpk) varies from 0.5h to 19.7h and the runoff concentration (VOL1h) varies between 3.2% and 30.1%. The TTD50 and TTDpk estimates of the events show a homogeneity by season of occurrence. Moreover, these estimates have a low uncertainty given the total variability observed over the year, except for the period from March to May.

KOE and HM exhibit similar TTD during the mid-October to mid-April period, although the hydrological transfer on HM is

almost constantly slightly quicker (-2h in average for TTD50) and slightly more concentrated (+2,5 % in average). In contrast, significant discrepancies between both catchment sections are observed during the summer period (mid-April to mid-October). For the HM section, the TTD50 decreases from an average of 8.9h in winter to half the value (4.6h) in summer. In contrast, the TTD50 shows less variability for the KOE catchment, and even an increase by 1.6h in summer, suggesting an opposite effect of the dry conditions on catchment responses. Eventually, TTD50 in summer is on average 2.6

times shorter for the HM section than for the KOE catchment. The peak lag times show even more contrasted values, with the average TTDpk during summer being 1.9h and 8.5h for the HM section and the KOE catchment respectively. We may also note the very high reactivity (i.e., short response time) of the HM section, considering its area.

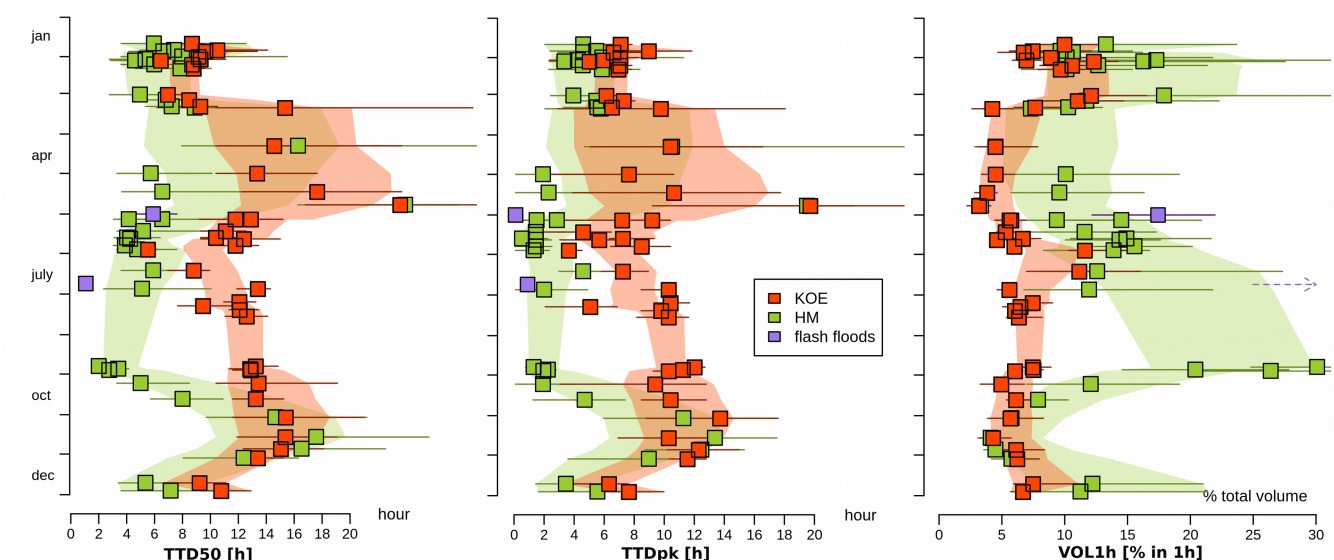

**Figure 9: Properties of the simulated transfer time distributions: the median transfer time (TTD50 [h], left panel ), the peak flow lag time (TTDpk [h], center), the runoff response concentration in one hour (VOL1h [%], right panel). The events are ordered by calendar day. The orange and green envelopes correspond to the average calendar values, based on the 3 closest estimates and taking into account the uncertainties of the metric (TTD50, TTDpk or VOL1h) assessments. The purple arrow on the third panel points to VOL1H of the flash flood of July 2016 which exceeds the graph scale (VOL1H = 64.3%).**

The TTD spread (VOL1h or runoff concentration) shows also different variations along the season, depending on the catchment considered. For the KOE catchment, VOL1h varies only moderately throughout the seasons around the small average of 7.1 % (σ = 2.3 %). A notable exception is the February-March period, when antecedent wetness is at its highest and VOL1h then reaches 9.4 % (σ = 2.0 %). Relatively high values of VOL1h also define the hydrological responses of the extreme event of 13[th] July 2021 and the high rainfall intensity event of the 26[th] June 2020. For the HM section, the assessment of VOL1h is highly uncertain, but a seasonal trend can nevertheless be identified: there are two periods of concentrated TTDs corresponding to the January-February months and the end of the summer period in September. In this later period the TTDs are particularly concentrated, with VOL1h varying between 20 and 30%. In contrast, transition periods, i.e., the recharging in autumn (end of October, November) and the drying out in spring (end of March, April) have the lowest VOL1h (7.2± 2.7%).

The 2016 and 2018 flash flood event TTDs (table 5) are partly in the lower range of variation for the hydrological responses of the HM section during summer, with the first one being significantly more concentrated (50%, outside the chart's limits) and the second one exhibiting a 6-minutes flood peak occurrence.

Finally, the TTD properties show that the Koedange subcatchment is much more resilient to rainfall variability and catchment water state, exhibiting less variability along the seasons, and reflecting damped and delayed hydrological responses. In contrast, the high variability of the HM's TTD highlights its non-linear response, and its specific sensitivity to soil wetness, storage levels and rainfall forcing. More specifically, this catchment section appears to be vulnerable to flash

flood processing as the hydrological response peak occurs really shortly after rainfall forcing and in a concentrated way during the summer period.

**Table 5: Seasonal average of the TTD properties.**

| | Koedange | | Medernach | | Flash flood events at Larochette | |
|---|---|---|---|---|---|---|
| | 16 Oct. – 14 April | 15 April – 15 Oct. | 16 Oct. – 14 April | 15 April – 15 Oct. | 22-07-2016 | 31-05-2018 |
| TTD50 [h] | 10.9 + 3.0 | 12.0 + 2.4 | 8.9+ 4.1 | 4.6 + 1.3 | 1.1 + 0.1 | 5.2 + 0.75 |
| TTDpk [h] | 8.3 ± 2.4 | 8.5 ± 2.4 | 6.5 ± 3.1 | 1.9 ± 0.9 | 0.9 ± 0.1 | 0.1 ± 0 |
| VOL1h [%] | 7.7 ± 2.5 | 6.5 ± 2.0 | 10.2 ± 4.1 | 15.1 ± 6.0 | 64.3 ± 5.0 | 17.4 ± 2.5 |

### 4.3 Relating the seasonal TTD variation to the rainfall forcing and the catchment wetness state

The correlation between the hydrological response properties (RC, TTD50, TTDpk, VOL1h), the catchment eco-hydrological state (Qbase, SWC50, SWC20, DAY), and the rainfall forcing properties (Rduration, Rcumul, I1h, I15min, Imean) are studied using Kendall's τ (Kendall, 1938) and Hoeffding's *D* (Hoeffding, 1948) correlation tests. Figure 10 illustrates the variation of the catchment state and of the rainfall properties proper to the events. Figures 11 and 12 show the Kendall's τ and Hoeffding's D correlation matrices for KOE (left panels) and HM (right panels), respectively.

For the KOE catchment, the properties of the hydrological responses show almost no significant correlation with the rainfall properties. Only the runoff coefficients appear to have a moderate non-monotonic correlation with rainfall duration. The transfer time distributions appear to be totally independent of the rainfall properties, except for the peak lag times that are weakly correlated to the maximum precipitation in 6 hours (I6h). In contrast, the TTDs properties of show dependencies on the catchment wetness state. More specifically, the runoff coefficient is highly correlated with all catchment properties

(SWC20, SWC50, Qbase).. The median transfer time (TTD50) and the TTD damping (VOL1h) – which are highly anticorrelated – are linked to the soil moisture states (SWC50 and SWC20) in a non-monotonic way. Note that the highest correlation to the transfer lag times (TTD50, TTDpk) is obtained with the seasonal period (DAY, non-monotonic correlation), which contrasts with the lack of correlation with the baseflow (Qbase).

We find slightly contrasted results for the HM section. As for the KOE catchment, the runoff coefficient is strongly linked to

the catchment wetness state and less to rainfall properties. However, the TTD variability shows an almost opposite correlation to the one observed for the KOE catchment. The TTD properties are correlated to 4 out of 5 of the studied rainfall properties. Specifically, the characteristic lag times (TTD50, TTDpk) are highly correlated with the mean rainfall intensity (Imean) and the maximum hourly rainfall rates (I1h). There is a moderate (or high non-monotonic) correlation between the transfer lag time (TTD50, TTDpk) and the catchment states (Qbase, SWC20, SWC50), but not at all with the seasonal period

(DAY). Note that moderate correlations between catchment states and rainfall properties appear in this catchment, which might confound the interdependencies observed in this analysis. Finally, the TTD spread appears to be moderately linked to the rainfall properties (Rduration and Imean, particularly).

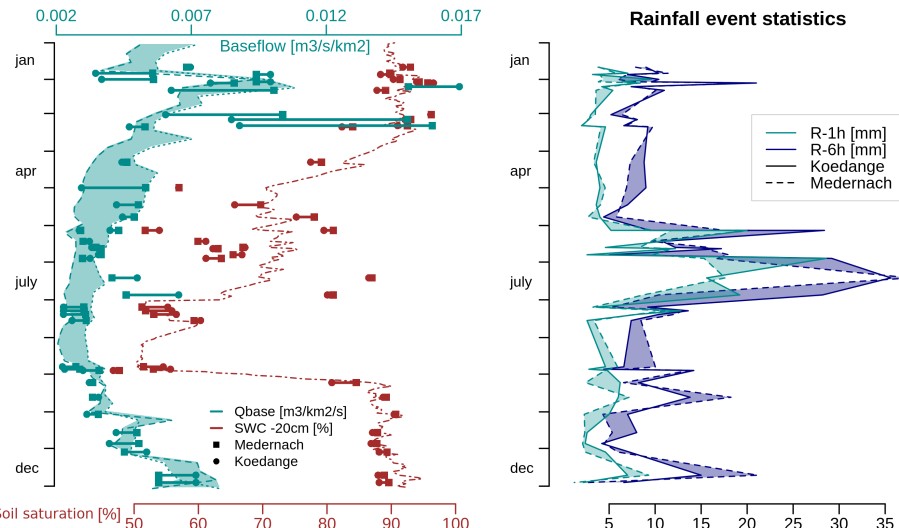

**Figure 10. Left: The catchment state at the start of each event (points): The minimum discharge during the 7 days before the event (Qbase, [m³.km⁻².s⁻¹]), and the soil saturation at 20cm in depth (SWC20 [%]). The light blue colour corresponds to the weekly average discharge minimum at Koedange (solid line) and Medernach (dashed line) over the studied period. The red line corresponds to the soil saturation calendar day average at 20 cm depth in the Medernach catchment over the same period. Right: The rainfall properties: the maximal hourly rainfall intensity (R-1h [mm], light blue), and the maximum rainfall amount over 6 hours (R-6h [mm], dark blue).**


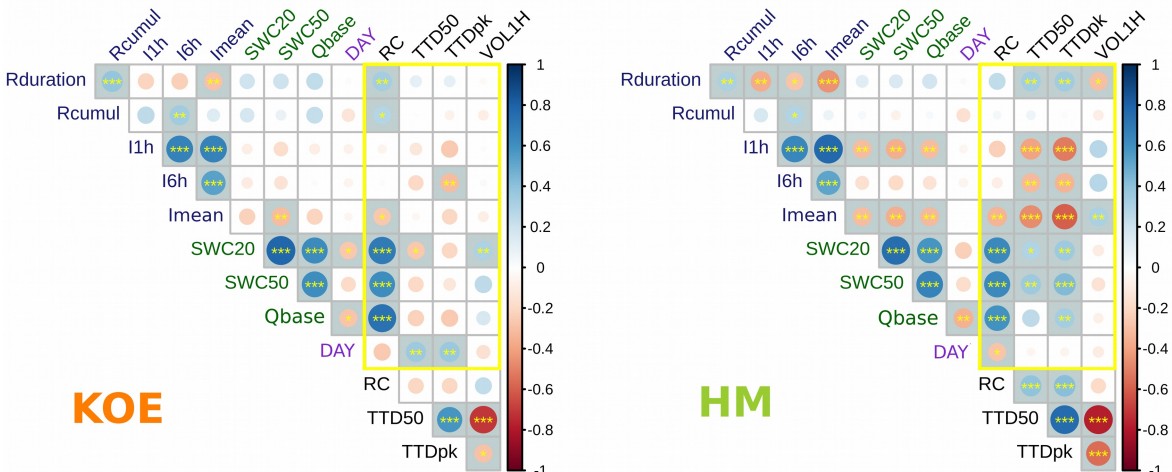

**Figure 11: Kendall correlation coefficients between rainfall (blue), catchment hydrological states (green), seasonal cycle (DAY of the year) and outlet runoff properties (black). See section 3.2 for more details on properties. The left and the right panels refer to KOE and HM catchment section respectively. The size and the color of the circles are related to the Kendall ceofficients. The yellow box highlights the scores of interest for our study. The blue background and the redstars indicate the significant correlations: *** when p-value < $10^{-3}$; ** when $10^{-3}$ < p-value < $10^{-2}$; * when $10^{-2}$ < p-value < $2.10^{-2}$.**

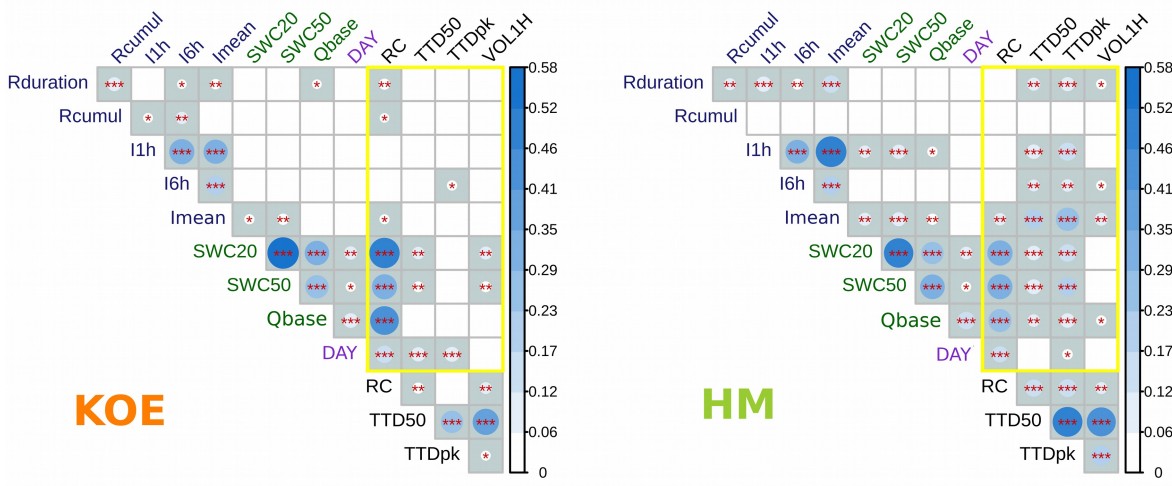

**Figure 12: Hoeffding correlation coefficients between rainfall (blue), catchment hydrological states (green), seasonal cycle (DAY of the year) and outlet runoff properties (black). See section 3.2 for more details on properties. The left and the right panels refer to KOE and HM catchment section respectively. The size and the color of the circles are related to the Hoeffding ceofficients. The yellow box highlights the scores of interest for our study. The blue background and the redstars indicate the significant correlations: *** when p-value < $10^{-3}$; ** when $10^{-3}$ < p-value < $10^{-2;}$ * when $10^{-2}$ < p-value < $2.10^{-2}$.**

## 5 Discussion

In our set of nested catchments with contrasted physiographic characteristics, we have targeted a better understanding of runoff generation processes during flash floods – and more specifically their respective timing. The catchment has been

extensively instrumented for differentiating the hydrological responses of several catchment sections. We studied two sections of similar dimensions and routing distance distributions, but with different substrate and structure. The KOE catchment has a marly substrate (Km3) and moderately steep Luxembourg sandstone outcrops (Li2). The HM section has its

drainage network deeply cut into the Luxembourg sandstone, with the latter being half covered by marly plateaus (Li3) with heavy clay soil. We applied a unit hydrograph model to properly extract comparable transfer time distributions of the net rainfall from the hillside to the outlet of both catchment sections. Both TTD sets relating to the 2019-2021 rainfall-runoff event database are compared and linked to the catchment hydrological state and rainfall properties.

## 5.1 Insights gained on model assumptions and limitations

The application of the unit hydrograph model has revealed its limitations for simulating some specific rainfall-runoff events in specific catchment sections. These limitations can be linked to the assumptions that the model relies on. This may eventually give us a hint to the actual mechanisms and hydrological functioning of both catchment sections.

Under wet but not yet saturated conditions, the model overestimates the discharge during the rising limb of the flood wave for the KOE catchment, while it underestimates and delays the flood peak. This suggests the actual net rainfall to be rather

small at the start of the event and larger towards the end. Additional simulations on the KOE catchment with lower RC during the first 20 hours of each event (table S4 and Figure S5 on supplementary materials) support this conjecture - the FDC simulation scores being slightly better for 11 out of 40 rainfall-runoff events (mostly occurring during the November-Mai period). It is likely that the first rainfall amounts reactivate the water paths to the river, resulting in a low RC at the beginning of the event that gradually increases towards a nominal value. Nevertheless, the simulations carried out with a

variable RC show little impact on the assessments of the TTD properties, except for a decrease in the confidence intervals for the April-May period. They also lead to the same seasonal variation already observed and described with constant RC. For the HM section, the limitation of a constant RC appears to be less critical. But rather than suggesting a difference in catchment behaviours, this finding is probably linked to the fact that the unit hydrograph model is only a part of the entire discharge simulation (with the other part – i.e., the hydraulic transfer, being well simulated).

For the KOE catchment, the flood peak of the highest 1h-rainfall intensity event (26/06/2020) is underestimated. One explanation can be that the infiltration capacity has been exceeded during the short period of intensive rainfall (I-1hour = 17.2 mm.h$^{-1}$). Assuming a steady RC for the entire event was again not appropriate for calculating the net rainfall distribution. The peculiar TTD of this event in comparison to the other summer events corroborates a change in the partitioning of the involved hydrological processes (faster overflow, resulting in a quicker response for this event).

For the HM section, we noticed that for high intensity events, the almost instantaneous and furtive flood peaks are not well simulated. Here, we propose two non-excluding mechanisms:

- As for the KOE catchment, the infiltration capacity has been reached, causing the net rainfall to be underestimated during the time steps with high intensity rainfall. In contrast to the KOE catchment, this is the case for several events and not only for particularly high rainfall intensities. This finding suggests an overall lower infiltration

capacity, which is in full agreement with the lower permeability that characterizes the clay soils of the marly plateaus. Also, the sensitivity of the TTD properties to rainfall characteristics reported in IV.3. corroborates this interpretation.

- The erratic three-peaked response observed after the impulse-like forcing of the 12.06.2020 event (Figure 8-c) highlights the spatial heterogeneity of the water transfer to the outlet. The low dispersion of the three peaks suggests
distinct and quick flow paths, almost without damping or buffering effects on the rainfall distribution. Rather than different flow paths in a same vertical profile, it is more likely that the different flow paths correspond to different tributaries that first concentrated and routed the net rainfall. The unit hydrograph model failed here to simulate a rather complex response, as the gamma function hinted that the soil and/or the substrate would get little but enough dampening effects to inhibit the impact of the stream network layout. The HM section's behaviour was eventually
similar to that of an urban or paved area.

## 5.2 Conjectured hydrological processes in the studied catchment sections

### 5.2.1 The KOE catchment

For the KOE catchment, the runoff transfer time shows little variability, which nevertheless delimits four periods: in winter (Jan-Feb) the observed transfer times are the shortest, followed by an abrupt transition to spring with the longest transfer
times (March-May). The transfer lag times get to a minimum at the end of June and July, before again increasing until the beginning of autumn (end of October, November). This double variation over the hydrological year suggests complex interactions, since all the assumed influencing variables (catchment water states and rainfall properties) are characterised by monotonic variation (only one increasing and one decreasing period for each variable) along the hydrological year. The high non-monotonic correlation with the calendar day suggests a stable variability along the study period although the first year
was rather dry and the second year rather wet. The stable seasonality suggests a hydrological functioning related to groundwater or deep layer interflow processes which are less impacted by inter-annual variability in comparison to runoff. More particularly, a possible interpretation is the buffering potential of the flat area around the river network which reacts more or less rapidly depending on its saturation state. The fact that the TTD properties are not at all or only moderately correlated with Qbase and SWC50 respectively may appear to be contradictory. In fact, it is rather that the indices are not as
representative as expected. Qbase, defined as the minimum flow over 7 days prior to the event, shows a high variability in winter that is not very representative of the gradual filling of the river's water table. Likewise, soil moisture measurements are obtained in the upper sections of the catchment, causing a significant time lag or difference with the soil moisture content of the bottom valley – the latter reflecting more clearly the hydrological connectivity of the flat area near the river.

The limited influence of the rainfall properties suggest that the critical zone is resilient to the climatic forcing, and that it
enables important vertical infiltration (and water storage), which has been only exceeded during one event. The moderate correlation between the peak flow lag time and the 6 hours rainfall amount nevertheless suggests a light impact which might

explain the complex seasonal variation, or the transfer lag times variability in June and July with lowest values with the highest rainfall intensity events.

The hydrological processes suggested here can be compared to those found in the Wollefsbach catchment (4.5 km$^2$) in Luxembourg (Wrede et al., 2015; Fenicia et al., 2014). This almost 100% marly (km3) catchment has a rather large storage capacity, despite the limited permeability of its underlying bedrock. The concept of variable contributing areas, according to soil and deep layer connectivity (wetness), is also suggested to explain the seasonality in hydrological responses. Fenicia et al. (2014) eventually found that the serial reservoir model is better suited for simulating the hydrological behaviour of the catchment, which has been justified by the fact that flows are predominantly lateral. The similarities between the Koedange and Wollefsbach catchments eventually concur for suggesting the main role of the flat marly terrain in the vicinity of the river that covers half of the downstream part of the Koedange subcatchment (table 1).

### 5.2.2 The HM section

For the HM section, the runoff transfer time exhibits high variability throughout the year, highlighting the influence of the climate forcing and environmental states on the hydrological processes. As for KOE the double variations (two increases and decreases) within a hydrological year suggests complex influences of the various compartments of the Critical Zone.

In the HM section, the longest lag times are observed in November, when soil wetness is still moderate. As the soil wetness increases through winter, the lag times gradually decrease – suggesting the onset of subsurface hydrological connectivity, similar to that observed for the KOE catchment. Note that both catchments exhibit a similar variability in their lag times throughout the winter and spring periods, but they significantly differ from May to October (Figure 9). The lag times tend to rapidly decrease in May, alongside a concentration of discharge volumes around (almost instantaneously occurring) peak flows, reaching their lowest values by end of September, early October. These substantial changes in the hydrological response suggest the onset of different processes, compared to the winter season. Note that RC are one order of magnitude smaller in summer than in winter, equally suggesting a major shift in the dominating hydrological processes – corresponding in summer to the onset of surface and sub-surface contributions.

The hydrological behaviour of the HM section has similarities to those observed in catchments generating two peak hydrographs. In this type of catchment, a first fast peak is commonly assumed to be generated either through saturation-excess overland flow in near-stream areas (e.g., Kirnbauer et al., 2005; Westhoff et al., 2011; Padilla et al., 2015; Martinez-Carreras et al., 2016), or via fast subsurface flow through macropores or fractures along the hillslopes (Jackisch et al., 2016; Martinez-Carreras et al., 2016; Gabrielli et al., 2012). The delayed second peak is commonly linked to groundwater processes, e.g., through a piston effect and/or an increasing connectivity to the riparian zone with the rise of GW levels and/or soil saturation (Onda et al., 2006).

**5.3 Are the conjectured hydrological processes on the basis of moderate events analysis transferable to extreme events ?**

An important hypothesis of our study is that the high responsiveness of a catchment can be detected from moderate rainfall events, which means that there is no threshold effect between the intensity of precipitation and the reactivity of a watershed. From the moderate events that we have analysed, we have evidence that the HM basin is more likely to generate fast floods because its hydrological response is about twice as fast and more concentrated in summer (in comparison to the KOE catchment). A first element supporting our hypothesis is the fact that the correlation analysis shows very little dependence of the hydrological responsiveness of the KOE catchment on general precipitation properties. This shows its resilience to precipitation characteristics, and thus a constancy in terms of responsiveness. Note that this statement only holds within the range of variation of the analysed precipitation properties and a possible threshold effect beyond this range of variation cannot be completely excluded. However, three events in the database include precipitation intensities of more than 15 mm in one hour. Without being extreme, this is close to the properties of the flash flood events reported in Table 2 and Figure 4. A closer analysis of the hydrological responses of two of these three events with high intensities shows that the response times of the KOE catchment are relatively shorter than those observed in the summer period: the median TTDpk are 3.75h and 7.25h and the VOL1H are 11.6% and 11.1% for the events of June 23, 2020 and July 13, 2021, respectively (in comparison to an average of 8.5h and 6.7% over the mid-April - mid-October period). It is thus possible that the correlation analysis via Kendall's indices may miss this dependence, by the fact that the strong main dependence on seasonality hides a minor dependence. Furthermore, these events are at the margin of those studied and Kendall's coefficients tend to minimise the influences of specific individual events. Assuming then that this influence is possible (despite the fact that it does not appear within the correlation analysis), we can still compare these response times and concentration rates to those observed for the HM catchment: median TTDpk are 1.30h and 2.0h and VOL1H is 13.9% and 11.9% for the events of 23 June 2020 and 13 July 2021, respectively. Thus, the response of HM remains both more concentrated, and above all more than twice as fast. The HM catchment still appears to be more prone to rapid/flash floods than the KOE catchment. Finally, if we compare the response times observed during flash floods in 2016 and 2018 (TTDpk = 0.1h and 0.9h; TTD50 = 5.2h and 1.1h, respectively), they are equivalent to the lowest response times observed in the HM catchment (TTDpk = 0.5h and TTD50 = 2h), supporting the fact that the high rainfall intensities of flash floods did not unequivocally generate faster runoff than moderate events (although the magnitude is not mentioned). Based on the correlation analysis and on the most intense summer events of our data set, we tend to conclude that the high responsiveness of the HM catchment (in comparison to the KOE catchment) prevails during more intense rainfall events and therefore corroborates our initial hypothesis.

### 5.4 Specificities of the HM section's onset of quick transfer runoff during dry summer conditions

Since our dataset appears to be (too) limited for validating our hypothesis, we propose here a list of plausible explanations – based on examples from scientific literature – for the drastic decrease in response times observed in summer on the HM
section, as opposed to the KOE section.

### 5.4.1 Why is there a quick transfer runoff on the HM section but not in the KOE catchment?

In the HM section a single fast peak response to rainfall is characteristic of the mid-April to mid-October period. The absent - or invisible - delayed groundwater response can be related to the unsaturated soil wetness that prevents any deep infiltration below the plateau, similarly to what has been observed by Martinez-Carreras et al. (2016) in a catchment with similar
landscape units. Note that we cannot conclude on an absence of a flat delayed response, similar to that observed for the KOE catchment, as the consecutive overlap of the two catchment responses cannot be distinguished due to the uncertainties in discharge measurements at these low water levels.

Previous studies have shown that the organization and distribution of landscape units can control the differences in runoff responses between nested catchments (Sidle et al., 2000; McGlynn et al., 2004; Iwasaki et al., 2020). Iwasaki et al. (2020)
studied 5 catchments with similar geology, climate and vegetation, but different geomorphological layout, and concluded on the key role of the riparian area in buffering fast hillslope flow mechanisms.

We conjecture that the contrasted hydrological response during summer in the KOE catchment - exhibiting no change in dominant hydrological processes - could be caused by:

- the larger riparian zone and the gentle slopes in the downstream part of the catchment, buffering the inflow of quick
runoff (Iwasaki et al., 2015; Iwasaki et al., 2020). In support of the role of the riparian zone as a buffer to rapid flow, we notice the same hydrograph patterns (Figure 5) at the following hydrometric station (Heffingen, Figure 1), suggesting similar hydrological processes, whereas a river restoration project has been carried out along the hydraulic section between Koedange and Heffingen to improve the lateral connectivity with the major river bed. In contrast, the riparian zone of the HM section is narrower and more urbanized, which further limits the presence of
buffer zones for surface runoff, such as wetlands.
- the less fractured Luxembourg sandstone in the KOE catchment might be less prone to trigger rapid flow paths contributions to the river. Highly fractured substrates can indeed serve as preferential pathways for significant subsurface flows (Graham et al., 2010). Focusing on hillslope processes, Gabrielli et al. (2012) similarly showed the key role of the weathered substrate layers in the setting up of preferential lateral flow paths during storm events in
the Maimai catchment (New Zealand).

### 5.4.2 Why is there quick transfer runoff on HM in summer and not during winter?

Note that in principle quick transfers of water might also occur in winter in the HM section, albeit mostly hidden by larger groundwater contributions. However, a detailed scrutineering of the hydrographs did not reveal any intermediate peak flows during the rising limb of the flood hydrographs, that could have supported this conjecture. Consequently, we conclude that the summer conditions are particularly prone to fast flow paths.

**The impact of dry conditions:**

Several studies, focusing on subsurface flow celerity on hillslopes, assessed the quicker flows during dry conditions (Scaini et al., 2018; Anderson et al., 2009; Asano et al., 2020), although they could not identify correlations between hillslope flow celerity and antecedent wetness conditions (Scaini et al., 2018; Iwasaki et al., 2020). The dry conditions are characterised by a large variability in hillslope responses (both in terms of volumes and timing), which decrease during wetter conditions (Scaini et al., 2018; Bergstrom et al., 2016; Teschemacher et al., 2019). The latter observation could explain the difficulty to assess correlations with highly variable celerity at the hillslope scale.

In our study, we observed moderate or highly non-monotonic correlations between response times and wetness states. In particular, the shortest response times were recorded in September when the soil moisture levels were lowest (Figure 9 and 10) and just before the soil re-wetting in October. In contrast to KOE, the soil moisture measurements in the top clay plateaus seem to be representative of the response times and therefore a plausible actor. Moreover, the absence of correlation with the seasonal variability (DAY) supports the hypothesis that it is precisely the moisture conditions inherent to previous rainfall events that are of importance, thus supporting the relevance of surface and subsurface moisture conditions on fast runoff transfer.

**The impact of the hydrophobic properties of the land surface:**

It can be assumed that dry conditions strongly limit the infiltration capability of soils, especially of clay soils, and ultimately support the onset of rapid surface processes. But this does not explain the runoff on sandy soil (with a high theoretical infiltration capacity) that was observed on sloping grassland during the 12/06/2020 event. We therefore conjecture that the hydrophobic characteristics of the soil surface, prevent runoff from being slowed down or retained as it travels downslope. The hydrological network of the HM section is mostly surrounded by steep and forested hillslopes, the latter exhibiting a pronounced seasonality in forest litter properties, including lateral permeability and hydrophobic behaviour during dry conditions.

Prior studies have shown that organic litter can contribute to the onset of subsurface flows – also known as biomat flows (Sato et al., 2004; Sidle et al., 2007; Kim et al., 2014; Du et al., 2019). Forest litter (especially under deciduous trees) develops a lateral structure due to the incremental horizontal accumulation of leaves or needles. At plot scale, Sidle et al., (2007) and Du et al., (2019) showed that the biomat flow can reach up to 44.6% (46.3%) and 12.3% (28.5%) of the total precipitation in pines and forest litter respectively, which was roughly three to eight times larger than Hortonian flow.

Also, in addition to the lateral structure, the litter - which is particularly rich in organic matter - can develop hydrophobic properties under dry conditions (Zavala et al., 2009; Kim et al., 2014) and consequently inhibit infiltration and promote runoff (Doerr et al., 2000; Gomi et al., 2008; Gerke et al., 2015; Jeyakumar et al., 2014). For example, Miyata et al. (2009) have shown that the soil water repellency enhanced the occurrence of pseudo-surface runoff during dry conditions. Although the factors controlling the hydrophobic property are not yet fully understood, the soil/litter moisture has been conjectured to be a key factor (Doerr et al., 2000; Butzen et al., 2015). Therefore, we can assume that the influence of forest floor water repellency on hydrological processes is largely seasonal.

Thus, despite the highly permeable sandy soils that cover the steep hillslopes of the HM sector, the infiltration capacity may be limited at times by the properties of the (forest) ground cover during dry conditions. The steep slopes could then potentially develop quick flow paths, eventually rapidly connected to the main river. Furthermore, the factual observation of surface runoff on an open slope (in grassland) during the 12/06/2020 event, leads us to generalize the key role of hydrophobic or infiltration properties of soil surface on steep slopes.

## 6. Conclusion

We analysed the runoff transfer time distribution over a complete year in catchments that have been recently affected by flash floods. The two studied catchments have similar size, elevation ranges and slopes, but differ in terms of geological substrates and landscape features.

While the variability in runoff coefficients is explained for both catchments by the soil storage dynamics, the variability in TTD has different causes. In the KOE catchment, the water transfer exhibits a seasonal variation, disconnected from precipitation characteristics (except for 4 summer events).

The HM section exhibits contrasted TTDs throughout the year, suggesting threshold dependent hydrological processes. More specifically, quick runoff transfers seem to dominate under dry conditions. Particularly the median transfer time and the peak lag time decrease 2 and 3 times respectively between the mid-April – mid-October period and the remaining part of the year. We conjecture that the rapid flows in the HM section are not only triggered on and by its marly plateaus, but also by the hydrophobic forest litter and soil cover of the sloping hillsides during dry conditions. The topographical connectivity of the steep slopes could develop flowpaths prone to a rapid transfer of water. The absence of a riparian zone prevents any dampening of these abrupt and massive flows in the case of extreme precipitation events.

When targeting an improvement in flash flood forecasting in Luxembourg, our results suggest that the focus should be set on the development of a simulation tool adapted to catchments with physiographic characteristics similar to those of the HM sub-catchment – i.e., with fractured bedrock and limited riparian zones. The non-linear hydrological behaviour of the basin throughout the seasons requires either the implementation of a complex model that considers the non-monotone relation between transfer velocity and soil wetness, or the set-up of a simpler model with a seasonal calibration.

In general, catchments with little or no dampening zones and steep slopes require specific attention and more focused investigations on flash flood generation processes.

More research is needed on the onset and role of infiltration processes, as well as surface and sub-surface flows at hillslope scale, under dry conditions. The latter may lead to limited infiltration capacities on the marly plateaus, while triggering at the same time the onset of surface flows on steep forested slopes. These investigations will have to combine multiple spatial
(i.e., plots, hillslopes, catchments) and temporal scales (from event to seasonal scale).

**Code availability**

The codes implementing the unit hydrograph model and the hydraulic transfer model can be found at the LIST GitLab (https://github.com/adouinot/TransitTimeModel, last access: 17-dec-2021).

**Data availability**

The rainfall and the discharge time series in this study are the property of the Luxembourg Institute of Science and Technology (LIST) and can be obtained upon request from cyrille.tailliez@list.lu, after approval by LIST.

**Author contribution**

AD and LP designed the project and obtained the funding for this study. AD, JFI and CT established the experimental set up of the monitoring network and the discharge measurements. AD performed the modelling part and carried out the correlation
analysis. AD, and LP jointly structured the paper, with contributions on interpretations of results from CM and JFI.

**Competing interests**

The authors declare that they have no conflict of interest.

**Acknowledgements**

This research has been carried out in the framework of a national public-private partnership funded by the National Research
Fund of Luxembourg (FNR AFR PPP grant 118823575), involving POST Telecom, the 'Administration de la Gestion de l'Eau' (AGE), and the Luxembourg Institute of Science and Technology (LIST). We thank POST for providing the communication technology hardware, as well as their support in the design and implementation of the experimental set-up. We thank the AGE for their support in coordinating the project, their continuous and highly valuable input during project

meetings, as well as their support on securing funding through the 'Fonds pour la gestion de l'eau du Gouvernement de

Luxembourg' for the acquisition of project-specific monitoring devices.

**Financial support**

This research has been supported by the 'Fonds National de la Recherche du Luxembourg' (AFR PPP grant, n° 11823575), as well as through the 'Fonds pour la gestion de l'eau' of the government of Luxembourg.

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
