# Peer review of "Flood patterns in a catchment with mixed bedrock geology and a hilly landscape: identification of flashy runoff contributions during storm events"

_Hydrology and Earth System Sciences, 2022_

## Author Comment (AC1)

This study compares storm runoff processes between two catchments with similar size based on the analysis of transfer time distributions (TTDs). The authors present a seasonality in TTDs, which had a different trend between the two catchments. Quick runoff transfers occurred under dry condition in a catchment. The authors attribute the rapid flows to marly plateaus, hydrophobic forest litter, and the absence of a riparian zone in the catchment.

This paper deals with an important topic. I think their analysis of TTDs using a unit hydrograph model is effective for comparing the storm runoff characteristics between the neighboring catchments. Seasonality in TTDs (Figure 8) is especially interesting. However, data for discussing the causes of the seasonality and inter-catchment differences in TTDs are insufficient. More information about groundwater dynamics and topographic analysis is needed to discuss the causes of rapid flows. My major concerns are listed below, followed a list of specific comments and technical corrections.

General comments

Which is the novelty of this study, analytical methodology or the estimated causes of rapid runoff? If the TTD-based comparison of storm runoff characteristics is a novel approach, the authors should emphasize this content in Introduction and Discussion. If they think the causes of rapid runoff are the main findings, they should increase reliability of estimating the causes. In this manuscript, relationships between runoff mechanisms and TTDs are unclear and the factors causing rapid flows in their study sites are only speculated from the results in TTDs.

The TTD-based comparison is not a novel approach as it is already used either in old studies comparing the shapes of the hydrograph to classify a set of catchments or more recently to study the impact of catchment management (before and after restoration management, as example see Memberu et al, 2018, figure 6). From our point of view, the novelty presented here is a clear demonstration of the distinct impact of dry conditions on runoff transfer processes. It is thus shown that beyond the precipitation characteristics, the configuration of the catchment will be a key factor in the generation of a flash flood following a summer storm. In order to emphasize this idea, we reworded the paragraph summarizing the results:

*"We observed a seasonality of the TTDs for both catchments, with dry conditions having an opposite impact on them. The KOE catchment reacts less quickly and more spread out under dry conditions. On HM catchment on the contrary, response times are significantly shorter and concentrated (-59% ± 33%) and (+33% ± 87%).* **This opposite seasonality** *leads us to consider/hypothesize different control factors of the runoff transfer processes in relation with the topographic and geological layout of the catchment areas."*

(REF:Menberu, M. W., Haghighi, A. T., Ronkanen, A.-K., Marttila, H., & Kløve, B. (2018). Effects of Drainage and Subsequent Restoration on Peatland Hydrological Processes at Catchment Scale. Water Resources Research, 54(7), 4479–4497. doi.org/10.1029/2017WR022362)

Although the authors focus on bedrock geology, groundwater dynamics in each geology are unclear. The rapid flow due to hydrophobic forest litter was also not observed in their study sites. Moreover, differences in riparian topography between the catchments were not presented despite mentioning riparian buffering. Due to lack of these data, they only speculate the causes of rapid flow. If they want to discuss the causes based on the data in these two catchments, more detailed presentation of groundwater flows and topographic characteristics in the catchments is needed.

Unfortunately we do not have groundwater flows data, nor hydrophobic forest litter measurements. The main idea of the paper was to highlight the distinct behavior along the Ernz Blanche catchment on dry conditions. We orientating the discussion toward the causes because it seemed to us coherent to enumerate the possible factors - even though we cannot justify otherwise than by the literature. The idea here is to propose several avenues for further investigation.

Concerning the topography, it is true that we have reduced its presentation while it occupies a major part of our conclusion/discussion. We have therefore added some elements in the presentation section of the watersheds, in particular the iso-contours on figure 1 and a map superimposing slopes and Heights Above Nearest Drainage (HAND) highlighting the riparian zone on KOE and the steep slopes on the perimeter of the hydrographic network on HM.

I can't understand why the authors compared only two catchments despite the observations in six nested catchments (Figure 1). How different were TTDs between the six catchments? I think examination of relationships between TTDs and catchment characteristics (including geology, topography, catchment size, and vegetation) using the data of six catchments can provide more valuable implications. Even though groundwater flows were not observed, the causes of rapid flow may be estimated with reliability if the comparison of six catchments is conducted.

We carried out the analyses on only two catchment areas and not on all six embedded catchment areas, because on the one hand the flow series at two stations (Reisdorf, Hessemillen) are uncertain due to the backflow of a confluence and a dam respectively, and on the other hand the comparison of transfer time distributions of catchments of different size seemed less convincing to us. We have nevertheless mentioned them because the shape of the hydrographs clearly indicates a break in behavior between what happens upstream of Heffingen and what happens downstream (Larochette, Médernach, Hessemillen, Reisdorf). This is particularly showed/visible on Figure S1 in the supplementary materials.
* * *
Specific comments

Title: If the main theme of this paper is causes of flashy runoff, the causes should be examined more deeply based on other groundwater and topographic data or the comparison of TTDs between more catchments.

We changed the title for: **„Flood patterns in a catchment with mixed bedrock geology and topography: highlighting the delimited flashy runoff contributions during storm events"**

The initial objective was to find the causes of flashy contributions. Although we believe that we have identified and highlighted when and where rapid contributions occur on the White Ernz, we recognize that we do not have enough evidence to clearly identify the causes.

L14-17: Although the geology of the catchments is well described, there is little information about their topographic characteristics. I want the authors to clearly present the difference in topography between the catchments.

As suggested, We gave more details about the topographic characteristics:

The upper catchment (KOE) is dominated by **a low land area (38% of the catchment is located less than 30 m above the river network) consisting in variagated** marly bedrock (Midle Keuper Km3) and moderately steep Luxembourg sandstone outcrops (Lower Liassic Li2). The lower catchment (HM) has its drainage network deeply cut into the Luxembourg sandstone, with half of it being covered by marly plateaus (Lower Liassic Li3, **located between 80 m and 100 m above the river network)** featuring heavy clay soil.

L27-29: These causes are only the speculation and remains hypotheses. As these hypotheses were not verified, this is inappropriate as the conclusion. It may also be possible that the quick runoff under dry condition was caused by direct precipitation on stream channel and/or rapid runoff from riparian zone. As the catchment got wetter, hillslope runoff with long transfer time may contribute to stream water, which can be a possible mechanism of longer TTDs in wet conditions.

It seems to us that we are defending the same hypothesis: dry conditions imply direct and rapid runoff to the river. The real question is why this has more impact on HM than on KOE. We thus clarify our hypothesis by assuming that: I) the impact of dry conditions is stronger on marly

plateaus; ii) the dry litter of sloping forests favors runoff rather than water retention and infiltration, and iii) the riparian zone when sufficiently wide allows a buffer effect on this direct runoff. We agree that those hypotheses not actually proven by experimental results, but they are suggested as such: line 30: "stand as **our main hypotheses** in this respect".

L58-60: Whereas the authors wrote "The numerous faults and cracks support quick water transfer through the weathered bedrock and explain fast hydrological responses" in this sentence, they also wrote "Less permeable bedrock will lead to ... smaller catchment mean transit times." in L80-81. Whether the weathered bedrock can contribute to fast responses (smaller transit times) or not?
...both : depending at which time scale we look at. At event scale, we observed fast flows, but at seasonal scale the baseflow release is low. „catchment mean transit times" in Pfister et al. 2017 actually refers to „baseflow mean transit times". I replaced it to clearly make the distinction.

L89: If the main problem of previous research is the lack of observations in extreme events, this should be clearly presented in Background section. The event magnitude should also be emphasized in the Results and Discussions.
You're true. Our study is based on moderate events and the event's magnitudes do not make the our study specific. The specificity of our study is to focus on what impacts the speed and amplitude of runoff processes beyond rainfall properties. The last studies in Luxembourg does not enables to answer this question. We changed the lines 90-95 to clarify this idea.
*"To date, all investigations focusing on rainfall-runoff transformation processes in the Luxembourg context have been limited to small experimental watersheds (< 5 km²) or dedicated to storage and catchment release. While these studies have substantially improved our understanding of physiographic controls on runoff generation, we still have poor knowledge of the processes leading to quick runoff on catchments with a genuine river network as in flash flood events."*

L98-99: I could not understand the difference in flash flood type between Central Europe and MA regions. Please describe the difference more clearly in Background.
The lines 74-78 explain the different context between Central Europe and MA region which makes impossible to transfer knowledge of the flash flood processes in MA to the Central Europe. We thought it is clear enough but we give more details here:
*"While most flash flood related literature published to date refers to the Mediterranean area (MA), the processes underlying flash floods in Central Europe remain poorly understood. This mainly relates to the fact that :*
- *in these catchments (i) the climate forcing is not primarily controlled by topography (as opposed to MA),* → in MA, the Alps, the Pyrenees, the Cévennes foothills consists in elevated mountains which can block and induce convective and stable (in space) storms. In Central Europe, there is no such topographical barrier. Although the 2018 and the 2016 flash flood event were induced by relatively high rainfall amount, this is not the same order of magnitude than in MA (~50 mm compared to ~200 mm)
- *(ii) catchment storage filling states are very different between early summer (storage levels being still high when flash floods occur in Central European catchments) and autumn (storage levels being low when flash floods occur in MA catchments)* → the catchment conditions are not the same as the season of occurrence is different so the conclusion found in MA could not be transfer to Central Europe.
- *and (iii) the underlying bedrock geology is very different between Central European and MA catchments."* → *As well, some studies on MA relate the flash flood processes to specific bedrock geology. Here in Central Europe, the bedrock is linked to other mountains formations, so the impact could not be deduced again.*

L116-162: I could not understand which catchments had more permeable bedrock and larger storage capacity. According to Table 1, geology seems similar between the catchments: Both had the main

geology of sandstone and second geology of marls. If the authors focus on the geological features, geological difference between the catchments should be explained more clearly. Information about vegetation is also required because the effects of litter are discussed.

At a first glance the catchment seems to get the same geology; half part being marls and the other part Luxembourg sandstones. However there are two big differences:

- First, the marls layers consists in two different geological substrates with different properties. On Koe catchment the marls is a middle Keuper stratum (km3, Trias superior) and on HM the marls date from the Sinemurian period (Li3, mars of Strassen). The middle Keuper marls actually includes conglomerates and thin beds of dolomite, which can in turns include aquifers that are significant enough to be mentioned (Bouezmarni et Debbaut, 2006). Conversely, the Strassen marl is revealed in the landscape by the appearance of numerous springs at its upper limit. These features tend to reveal a relative permeability of km3 in comparison to Li3.
- Second, the arrangement of the sandstones and the marls layers is reversed. On HM, the marls of Strassen consist in the plateau, i.e the top of the relief, while the midlle Keuper marls on KOE consists in the riparian zone. Because of it respective location, the marls layer on HM will be more sensitive to dry condition.

We make several changes in section 2.1 and table 1 to be more accurate on that point. When speaking of km3 marls, we said "variagated marls". On table 1, instead of speaking of the first and second geology in terms of size area, we speak about "lower geology" and "overhanging geology".

(REF: Bouezmani, Debbaut. Carte géologique de Wallonie, Tintigny, Etalle, Notice explicative. Université de Liège, 2006.)

Figure 1: Please add contour lines in the figure. Addition of the map of slope angle or topographic index is also helpful to understand the topographic features of study sites. As soil moisture was observed at the points of raingauges, "Raingauges" should be changed to a phrase such as "Raingauges and soil moisture observations".

The suggestions have been applied to improve the catchment's presentation: the topographic contour lines have been added to the figure and the legend has been modified. Furthermore a map which integrates the slope display and the heights above nearest drainage map. The latter information highlights the buffering area on KOE catchment and the closeness of the slopes to the main river network on HM catchment.

Table 1: What is the difference in river width and riparian area between the catchment sections? Similar area, elevation range, and slope range does not necessarily mean that the two catchments have similar topography.

We added a statistic about the height above nearest drainage that characterizes the difference in surface area of the riparian zone between both catchments.

L139: Does "deeply cut" mean that valley was deeper in HM section than KOE catchment? If so, this topographic characteristic should be quantitatively presented.

The surface area close to the river network in terms of elevation is 3 times smaller on HM catchment in comparison to the KOE one. This figure presented now in table 1 supports the qualification of "deeply cut" apply to the river network in HM catchment. In an illustrative way, the slope display in figure 1 now illustrates this characterization.

L212 "net rainfall amount after infiltration": How did you determine the amount of loss (i.e., total rainfall – net rainfall)?

This is based on the observation of the average runoff coefficient: the net rainfall volume is equal to the discharge volume observed. Assuming a constant RC along each event the net rainfall at each time $t_i$ is : $R_{net}(t_i) = RC* R_{total}(t_i)$  (this is already described in equation 7)

Figure 7: Hydrographs in Heffingen catchment were very clearly different from those in Koedange and Medernech catchments. Why was the runoff delayed in Heffingen? I think the comparison between various catchments may provide clearer insights into runoff mechanism than the comparison between only the two catchments.
See figure S1 in supplementary materials. The hydrographs in Heffingen catchments are different from the Medernach catchments but not from the Koedange catchment. With the latter there is only a delay and a spread, which is an expected behavior when looking at a downstream measurement. This has been already described lines 187-189 (first submission): *"The headwaters (as expressed through the Koedange and Heffingen stream gauges) consistently triggered rather attenuated hydrological responses. Further downstream, the stream gauges located downstream of Larochette exhibited a much more responsive behavioural pattern."*

Figure 8: Please add the results in runoff coefficients of each event. I also recommend the authors to conduct statistical in order to examine whether the difference in the TTD values between the two catchments was significant in each event.
We are not sure to understand your suggestion. The runoff coefficient is part of the observation and not calculated using the model. That's why it does not appear in the figure. Or do you mean to range event according to the runoff coefficients?

L418: I want the authors to show the location of "large flat terrain" in Figure 1 based on topographic map with the spatial distribution of slope angle or topographic index. It would also be helpful if the area of this flat terrain can be shown in Table 1.
This is now visible through the heights above nearest drainage statistics (table 1 and map (figure 1 right).

L421-422: Why does the limited permeability of underlying bedrock lead to large storage capacity? I think permeable bedrock has larger storage capacity because groundwater is stored within weathered layer or fractures in bedrock.
It is rather a misunderstanding of the connection between the two parts of the sentence. We reworded the sentence: *"This almost 100% marly (km3) catchment has a rather large storage capacity, **despite of**  the limited permeability of its underlying bedrock."*

L490-510: Although only the effect of litter layer is discussed, discussion about evapotranspiration is also necessary for the impact of the vegetation because LAI directly affects it.
As focusing on the time transfer distribution variability and not on the runoff coefficient ones, we do not discuss on evapotranspiration.

L513-514: Differences in geological substrates and landscape features between the catchments should be more clearly presented throughout the manuscript.
Thanks to your suggestion, we added several description that - we hope – will help the catchment's characterization.

L516-517: There is no evidence that main runoff source in the KOE catchment was groundwater and deep soil water.
We reworded this part of the conclusion*: " In the KOE catchment, the water transfer get a seasonal variation disconnected from precipitation characteristics  (except for one summer event)."*

L521-523, L529-530: It seems that the authors attribute the difference in runoff characteristics to topography in slope and riparian zone rather than geology. If so, stories focusing on the topographic features may be better.

As said before, changes have been made in line with this suggestion.

L531-532: Runoff coefficients were one of magnitude smaller in summer than in winter (L439). I think this result indicates that runoff during dry summer season had small risk of flooding even if the rapid flows occurred. Both results of runoff coefficients and TTDs should be considered to provide conclusion for flood risk management.

Our conclusion deals with the specific context of **flash** floods. In this context, the short timing and the peak magnitude rather – than the runoff coefficient – are of first importance. As a prove, the 2016 and 2018 **flash** flood events get not so high runoff coefficients (14% and 20%), while their flood peaks are among the three highest recorded.

→ ...
* * *
Technical corrections

L24: "Another catchment" would be better than "The HM section" because I could not understand this is the name of catchment when I firstly read Abstract.

The HM acronym presentation has been added line 18.

L100: Does "mean summer and winter runoff" mean baseflow runoff in summer and winter?

It means "runoff coefficient". This has been specified.

L111-114: I think these sentences are unnecessary. ok.

L154-156: The order of Figures 2 and 3 is reversed. The caption has been corrected.

L177: Although it was written that "the rainfall amount had to exceed 10 mm", there is an event with the rainfall amount of 9.8 mm (Table 2).

The threshold of 10 mm was actually applied on the raingauge observation average that make a slight difference with the rainfall amount on Medernach catchment (obtained with weights on raingauge according to the Thiessen polygons).

We changed the description: *"[…] according to the following criteria: i) the rainfall amount average on the 4 raingauges had to exceed 10 mm, [...]"*

Figures 6 and 7: Please check if the date of (c) is true. Were they really different between the two figures?

The dates of (6c, 7c) are right. The figure 6 shows the result of the KOE catchment simulation, and the figure 7 shows the same but for Medernach. On panel, We choose two different events, because it seems to us more illustrative of the model "weaknesses" on each catchment respectively.

L308-310: Were these values the ranges in both catchments?

Sorry but we do not understand the question.

Figure 10: The color of SWC20 and RC may be wrong.

This has been changed.

---

## Author Comment (AC2)

We thank you for your comments that helps us to improve the manuscript. Most of your concern is related to the confuse link of the study to the flash flood context and to the model limitation.

In this new manuscript, we first implemented another year of rainfall-runoff event to consolidate our results. It also allow us to challenge our methodology on the extrem rainfall-runoff of July 2021. Furtermore we included the 2016 and 2018 flash flood events in this new version, including statistics and applying the unit hydrograph on it. Results on moderate rainfall-runoff events could now be introduced against the flash flood context. We hope also, that some rewording will help you to see the connection of our study with this scope.

Concerning the limitation of the model, it is a deliberate choice to consider a very simple model that does not introduce any dependency hypothesis that we could not verify/measure. The objective of applying the model is to obtain a clean way of calculating comparable TTDs over the two catchments. The imperfect fit of the model introduces a bias and informs about the complexity of the hydrological responses, which in itself is already a result. What interests us more specifically is the variability of the TTDs, from one catchment to another, from one season to another. We have now verified by a sensitivity test that the modelling flaw only introduces a bias and does not change the observed variability. Therefore, we consider the model to be sufficient to calculate valid TTDs.

You will find the specific answers of your comments, with the modification applied to the manuscript.

**General comments:**

The occurrence of extreme events like flash floods are usually be linked to extraordinary catchment system states or precipitation characteristics. They may be a matter of threshold-behaviour. The analysis of the catchment runoff reaction on ordinary rainfall events with a linear model therefore does not necessarily contribute to the understanding of extreme events. Thus, I would strongly recommend to adapt the frame of your manuscript and agree with Prof. Dunkerley's comments.

We agree with you that precipitation properties play a strong role in the generation of flash floods. However, we also know - from the literature (Payrastre et al, 2013, Zanon et al, 2010 for example), that physiographic properties (relief, geology, pedology, land use) can play a key role in the acceleration of runoff processes, and the generation of very fast floods. This is what we study in this article: *What are the characteristics of a catchment that favours rapid and concentrated runoff transfers BEYOND the characteristics of rainfall*? We explore the Ernz Blanche catchment, which has been hit by record-breaking flash floods 3 times (1968, 2016, 2018), to identify what makes it particularly non-resilient to heavy rainfall. We have added the following lines to clarify our objective:

line 105: „– *what is influencing the specific flash flood event patterns, beyond the extrem rainfall properties?*"

In the context of flash floods, we believe that the study can help to identify among the ungauged catchments, those that favour rapid runoff processes.

By choosing to work on moderate events, we assume that the variability of catchment response times is also observed on moderate events. Figure 5 introduced in the new manuscript (and no longer in the supplementary materials) illustrates this assumption. We have added a line in the methodology to make it explicit:

line 127: *"We indeed assume that the hydrological reactivity of the catchment is detectable independently of the magnitude of the precipitation. "*

It is clear that extreme events are very difficult to measure and often do not occur in the short measurement periods available in projects. However, the rainfall-runoff dataset generated from

August 2019 to July 2020 is valuable for understanding the rainfall-runoff reaction of ordinary rainfall and maybe this should be the focus of the paper. In order to be able to better classify the measured events, the following information would be very helpful:

    a) information about the flash flood events in 2016 and 2018
    b) information about the probability of occurrence of the precipitation intensities. I am not aware whether information on design precipitation with defined return periods is available in Luxembourg. Even if such evaluations are of course subject to uncertainties, they can nevertheless provide guideline values for the classification of the measured precipitation events.
    c) Presumably there is a difference in the runoff response between long-lasting precipitation events and very short ones with high intensity. In order to work out these differences, it could be valuable to classify the precipitation events.

a) In line with your comment and those of Professor Dunkerley, we have included the flash flood events of 2016 and 2018 in the manuscript. We introduce their rainfall and runoff statistics in Table 2 and have applied the unit hydrograph model to them. Although the 2016 and 2018 rainfall and runoff measurements were recorded at different location (stream gauge at Larochette and not Medernach) and with different tool (rainfall radar measurement instead of the 4 raingauge network), they provide a benchmark to our rainfall-runoff event database.

b) Our raingauge network has-been installed in 2019. The discharge measurement at Larochette only start from 2014. This short measurement period make it not possible to get robust return period values. At Luxembourg scale there are 14 daily rainfall time series starting in the mid 50's. The hydrological network has been installed in the 90s, as well as subdaily rainfall measurements. Using these data, it would have been possible to obtain a return period value for precipitation. However, we preferred to present the events in the form of a principal component analysis. It gives a more encompassing picture of the events, as it includes other properties than those of the precipitation (flow, season of occurrence, soil moisture, …). In order to contextualise this picture, the flash flood events of 2016 and 218 have also been positioned on the graph.

c) We agree with the fact that there is a split between long winter events and their significant hydrological response in terms of volumes and the shorter and "smaller" summer events. Instead of a classification based on the duration of precipitation (which may not be adequate for some events), we preferred to rely on the PCA analysis which combines several rainfall properties and catchment states to reveal 2 rainfall-runoff event groups corresponding to two seasons: the October-April period and the May-September period (see Figure 4).

**Specific comments:**

Can you please justify in section 3.1. why you used a unit hydrograph model and why you assume a constant runoff coefficient? Even if there is no ideal model, you could also use another methodology, so it would be interesting for readers to know why you chose this one, which has some weaknesses pointed out in the discussion (e.g. line 385, 392).

The idea of working with a unit hydrograph model stems from the fact that we wanted to work with a real black box, making as few assumptions as possible about the hydrological functioning. We have an input signal (the rainfall), We have an output signal (the flow). We apply a transformation function (the gamma function) to go from one to the other. The choice of the transformation function does not imply any hydrological assumption, i.e. that none of the model parameters depend on the soil type, rainfall intensity, etc. This dependence should only be revealed by the correlation analysis in a second step. In other words, we did not introduce any hypothesis into the model that we wanted to verify later.

However, as you mentionned it, there is a strong assumption in the model that the runoff transfer is the same at all times, i.e. that the watershed system is in a steady state. There is actually a transitional phase where the runoff coefficient varies, as you mentioned, from a zero value to a nominal value, but also the transfer times must very likely be longer at the beginning than at the end of the event. The choice is made here to ignore this transitory phase for lack of being able to introduce it without making assumptions on the hydrological processes. In some ways this approach is very similar in assumption to a calculation of median transfer time as being equal to median runoff time minus median rainfall time.

To clarify our purposes we reworded the beginning of the section 3.1:

„We applied a simple unit hydrograph model to reproduce the hydrological responses of each rainfall forcing over each catchment section. The unit hydrograph model assumes (by definition) that each net rainfall unit has the same TTD. We assume that the runoff coefficient (RC) is constant during the event, and we thus consider our catchment in steady state. This strong assumption prevents us from imposing a transient phase (variable RC and TTD) that we cannot measure."

In line 395 you state that for high intensity events, flood peaks are not well simulated. This again shows the problem that your analysis is not so well embedded in the topic of flash floods.

More complex models with additional parameters will logically give a better reproduction of the hydrological response. However, we could not verify whether the better results will be for the right reasons (i.e. that the new assumptions behind the complex model are true). That's why we prefer to use a simple unit hydrograph model, and clearly identified in the manuscript the limitations of the model, being aware of the simplistic view of the catchment that we impose. We now have checked thank to your comments, that this limitation does not impact the assessement of the TTD properties, on which we base our study (see below).

Concerning the difficulties of the model to model the flood peaks of some summer events on the HM section, the description of the June 12, 2020 event (Figure 8 and line 365 - 368, in the modified manuscript) is very important. It shows that the response of a precipitation peak is multiplied into 3 flood peaks. This can only be modelled by integrating a spatialization of the flows, or even a hydraulic model.

It should be remembered that the aim of the model application is to extract response time distributions, not to obtain the best possible model. The model fails to model the three flood peaks, but results in the modelling of a flood peak located in the average of the three flood peaks, so we can assume that the modelling result is sufficient to extract the TTDs.

Concerning the relatively s on HM section and : this has to be lokked

A sensitivity study with changed runoff coefficients, as mentioned by Prof. Dunkerley, would at least be very helpful.

We haven't made sensitive analysis before getting your comment and the one mentioned by Prof. Dunkerley. Without making it exhaustively, we tested on the KOE catchment – i.e the one that seems to be more affected by the constant RC assumption – a variable RC along the event. Two Runoff Coefficients – $RC^m$ and $RC^p$ – have been defined: the first one characterizes the re-invigoration of the soil drainage at the beginning of the events and the second one characterizes the hydrological response in the heart of the flood respectively. Arbitrarily, the re-invigoration period is fixed to 20 h. $RC^m$ and $RC^p$ are calculated as indicated below:

- $$\frac{RC^m}{RC^p} = SCW\,20$$

- $$RC^m = \frac{RC \cdot V^{tot}}{V^m + SCW\,20 \cdot V^p}$$

where $V^{tot}$ is the total rainfall amount, $V^m$ the rainfall amount occurring during the first 20 hours, and $V^p = V^{tot} - V^m$. The impact on the FDC scores are presented in the supplementary materials, table S3. The TTD properties resulting in variable RC are compared to the TTD properties with constant RC on figure S4.

According to the FDC score, there is indeed an improvement of the results with a mean decrease of 2%. More specifically the results are significantly better with a variable RC for 11 out of the 40 events. Those events occurs during the November-May period.

Considering the TTD properties, TTD50 and TTDpk decreases in average by 0.5 h and 0.4 h respectively. The decrease is homogeneous on the data set. A largest difference appears during the April-Mai period, resulting in smaller range of transfer lag times uncertainties. Nevertheless, the seasonal variations of the TTD properties can be similarly observable on both unit hydrograph model simulations. The comments about TTDs properties thus based on the simulations with the constant RC hypothesis are still valid.

We recognize that the simulation results could be improved taking into account variable RC, in terms of scores and absolute values. Nevertheless, we assume – according to the presented test, that the general TTD properties variability observed over the seasons (and which is the subject of the paper) is consistent.

Please comment on possible differences between the LAI survey period and the period of your data (e.g. land use changes). Please explain how you compressed the LAI data with a spatial resolution of 1 km2 into a value for the correlation analysis. Did you use the mean value over the catchment area?

We recognized that the use of LAI that we dispose as an indicator of any influence of the vegetation cycle was not appropriate. We then removed it and replace it by the calendar day (DAY) as an indicator of the seasonal state of the catchment.

Please explain which soil moisture values you used in the correlation analysis? Did you use the value from the station situated in the respective catchment? If so, there are two stations in the HM catchment? Did you use the mean of both stations?

This is specify line 190: „***The observed soil humidity measurements were weighted according to the cover rate of each soil texture to account for their spatial variability***.“

As example in the HM section, the soil texture distribution is:
- 40% of sandy soil which covers Luxembourg sandstone (Li2, figure 2),
- 35% of clay sol which covers marls of Strassen (Li3, figure 2)
- 25% of clay sol which covers variagated marls of midlle Keuper (Km3, figure2)

We applied those rates on the three related soil moisture sensors (only the one located in KOE catchment is then not used here).

The lithological abbreviations (Km3, Li2, Li3) are confusing if one does not know the context (and therefore cannot assign the numberings – e.g. what is Li1?). In Fig. 1, some of the lithological units have an abbreviation, some do not. This looks very inconsistent. Would it be possible to do without the abbreviations (Km3, Li2, Li3)? If not, it would be important to at least cite the geological map from which these designations originate. In any case, I would delete the lithological abbreviations from the abstract.

Km3 : third layer from midlle Keuper (Triassic) period

Li2, Li3 : second and third layer from Lias (Jurassic) period

We choose to maintain the abbreviation as they are widely used at national level. As you suggested, we clarified the legend, systematically introducing the geological abbreviation. Furthermore, we

sorted the substrates, from the youngest to the oldest, in order to be readable for the uninitiated. As example the marls plateaus correspond to the yougest substrate.

We added a reference on Figure 1 related to the geological properties of the area:

*„Figure 1: Ernz Blanche catchment (102 km$^2$). Discharge and rainfall monitoring network; Left: geological substrates (see Kausch & Maquil (2018) for more details). [...]"*

384: What do you mean with: "the model overestimates the rising limb of the flood wave" – is it the duration of the rising limb, its slope or something else?

We mean that the simulated discharge was higher than the observed one during the rising of the flood wave. We reworded the sentence to be understandable:

Line 476 – 477: *„the model overestimates the discharge during the rising limb of the flood wave for the KOE catchment, while it underestimates and delays the flood peak"*

Fig. 10 and 11: please explain the size of the circles

The size, as the color of the circles, are related to the Kendall or Hoeffding ceofficients. This is now specified in the legend.

504: Is „pseudo" necessary?

We report here the conclusions of another publication, and we have decided to keep the vocabulary proposed by the authors in order not to insert any interpretation of their result.

**Technical comments:**

We thank you for all the detailed corrections mentioned in this section. We have incorporated all of them, some of which are commented on below where other changes have been applied.

- "et al" should be "et al." in the whole paper (see https://www.hydrology-and-earth-system-sciences.net/submission.html#references)
- 15/17: delete "Km3, Li2, Li3" . We are keeping the information on geological substrates in the summary, as we think it is of primary importance. The full name is now preferred to be understandable to all.
- 16: add "(HM)" after lower catchment.
- 62/576: „Bronstert" (instead of „Bronstaert")
- 106: add "(TTD)" after transfer time distribution
- 110: delete "(TTD)" after transfer time distribution
- Fig 2 left and right are interchanged (Depending on whether the picture arrangement or the text is changed, the text in line 162 may have to be corrected.)
- Fig 3 light brown is not clearly visible, please use a darker colour
- 246, 322, 336, 347, 353: correct "VOL1H" to "VOL1h"
- 270: "one event" instead of "one events"
- Fig. 8: The labelling of the x-axes is wrong: it should be "median transfer time", "peak flow lag time", "runoff response concentration". You could also omit these words because they are explained in the figure caption and just write the abbreviations.Fig. 9: I cannot find an x-axis for reading the LAI-values.
- Fig. 10 upper line: SWC20 should be green, RC should be black
- 458: „McGlynn et al. 2004" instead of „McGlynn, McDonnell, Seibert, and Kendall 2004"

- 478: "Scaini et al. 2018" instead of "Scaini, et al. 2018"
- 500: "Hortonian flow" instead of "hortonian flow"
- You often use „note that", I would avoid this phrase (at least I would not use it so frequently), but this is a matter of taste. We reworded some sentence to decrease the use of this formulation.

References:

Olivier Payrastre, Eric Gaume, Pierre Javelle, Bruno Janet, Patrick Fourmigué, Philippe Lefort, André Martin, Brice Boudevillain, Pascal Brunet, Guy Delrieu, Lorenzo Marchi, Yoann Aubert, Elisabeth Dautrey, Laurence Durand, Michel Lang, Laurent Boissier, Johnny Douvinet, Claude Martin & l'équipe « enquêtes post-événements » d'HyMeX (2019) Hydrological analysis of the catastrophic flash flood of 15th June 2010 in the area of Draguignan (Var, France), La Houille Blanche, 105:3-4, 140-148, DOI: 10.1051/lhb/2019057

Zanon, F., Borga, M., Zoccatelli, D., Marchi, L., Gaume, E., Bonnifait, L., & Delrieu, G. (2010). Hydrological analysis of a flash flood across a climatic and geologic gradient: The September 18, 2007 event in Western Slovenia. Journal of Hydrology, 394(1), 182–197. https://doi.org/10.1016/j.jhydrol.2010.08.020

---

## Author Comment (AC3)

This paper presents an analysis of some high-quality rainfall and streamflow data collected in two adjacent catchments in Luxembourg (the Ernz Blanche basin). The objective, as reflected in the title of the manuscript, was to understand better the hydrological mechanisms resulting in flash flooding in this catchment. The paper is generally clear and straightforward to read, though I think that the main focus should have been more strongly on large rainfall events than on hydrologic response under more usual events.

We thank you for you review that was really helpful to improve our manuscript. Our main change according to your comments were:

- to include the 2016 and 2018 flash flood events in our study. The unit hydrograph model was applied on those two flash flood events. We can then connect our results on moderate rainfall-runoff events in the flash flood context. We hope also, that some rewording will help you to see the connection of our study with this scope.
- to insert a sensitivity analysis into the supplementary materials, in order to argue the small impact of our constant RC assumption on our results.
- Present more cautiously our explanations for the impact of the hydrophobicity of forest litter and the soil surface. We acknowledge that we do not have data to validate these explanations, but we believe that this open discussion could be beneficial to the scientific community, by opening up hypotheses to be tested.

In addition, we have taken advantage of this rewrite to add an event year to ensure the consistency of the correlation analyses and to work on two contrasting hydrological years (one being rather dry, the other rather wet).

Here below, we answer to each of your specific comments.

Oddly, though the authors mention the occurrence of several historical flash floods, including one in 2016 and another in 2018, they do not describe those events in any detail. They provide no discharge data, no runoff coefficients, and no rainfall event data. In order to find something of these events, I consulted an EGU Abstract (Iffly et al. 2018) by some of the same authors. There, I was able to learn that the 2016 event had much more intense rainfall than anything that the authors investigate in the present ms., recording 20 mm in 10 minutes (=120 mm/h), 50 mm in 1 hour, and up to 70 mm in 6 hours (=almost 12 mm/h). In contrast, in the present paper the most intense event reported had a maximum rainfall rate of ~ 27 mm/h. All but one of the remaining events listed in Table 2 had maximum intensities of < 10 mm/h. These seem unlikely to be responsible for flash floods. I was not able to locate information on the 2018 flash flood event for additional comparison. I think that it would help readers place the results of the current ms. in context, if some information on the historical flash floods could be provided, at least in summary.

As suggested, we added the 2016 and 2018 flash flood event properties (rainfall amount, flood peaks, runoff coefficients) in table 2, although the data is not exhaustive, with the highest impacts and flows of both floods being located downstream of the presented measurements. We also added a PCA analysis of the rainfall-runoff event dataset (Figure 4), on which the 2016 and 2018 flash flood statistics are positioned, which further contextualizes the study database in the context of flash floods.

I think that the focus on 'ordinary' events needed some comment. How can a study of much more ordinary rainfall events shed light on what occurs during the seemingly far more intense rainfalls that seemingly accounted for the historical flash floods?

We assume that there are not only the extreme properties of precipitation but the intrinsic catchment properties and its hydrological state that cause the hydrological response to be rapid and concentrated in a relatively large flood peak. In order words, we suppose that the catchment

"hydrological reactivity" is independent of the rainfall magnitude enough (although this will make the high hydrological reactivity to be problematic) to be detected on a moderate rainfall-runoff event database. In order to clarify this assumption, we added two sentences in the introduction:

lines 104: "*Here, we ask – in the context of a Central European study area – what is influencing the specific flash flood event patterns beyond the extrem rainfall properties?*"

line 123-124: "We indeed assume that the hydrological reactivity of the catchment is detectable independently of the magnitude of the precipitation. The same model is also applied on the 2016 and 2018 flash flood events, with the aim of having reference transfer times characteristic of flash floods."

Additionnaly we added one year of rainfall-runoff measurements to our study which results to enrich the database with 17 additional rainfall-runoff events. Among them, the event that occurred on the 13[th] July 2021 consists in an extrem event in terms of rainfall amount (129 mm) and discharge peak (the highest water level was recorded during that event since the oldest hydrometric station has been installed at Larochette in 2014). Although this event is not a flash flood, it enables to apply the TTD properties using extrem rainfall statistics, making possible to question the study result independency from the rainfall magnitude.

Did these [the flash floods], for instance, occur when the soil had been thoroughly wetted by antecedent rainfalls? Does surface runoff overtop ground surface roughness elements when the rain is sufficiently intense (above some threshold?), allowing a smoother and more direct path downslope? What was the nature of the precipitation? I assume that the flash floods were the result of shorter, more intense, convective events, and therefore were likely to have occurred in summer (this information is missing from the current ms.). I imagine that these were late afternoon events, but this would also be relevant information. Were there very local runoff sources located close to the stream channels, perhaps? Could the movement of convective cells parallel to the long, narrow catchment be significant? Did that occur (perhaps Doppler radar might shed some light on this)? Catchment response to intense convective cells might be quite different from that in stratiform rain, for instance, and different parts of the catchment might show altered hydrologic responses under those different rainfall inputs.

You are right, these are general characteristics of flash floods. But again (and perhaps this was poorly expressed), we are trying to determine what favours rapid and concentrated flooding, beyond the properties of rainstorms. the fact that they are convective events, of high intensity are recognized characteristics that favour flash floods, the fact that the rainstorm is located downstream of the catchment where the hydrographic network is strongly defined also. Here we seek to understand why two catchments react differently to the same rainfall event, whether it is intense or not.

Iffly et al. 2018 refer to lag times to runoff peak of just 90 minutes, whilst in the present study these lags extend to many hours.

In this study, the runoff peak response (TTDpk) varies from 0.5h to 13.7h, and more specifically on HM catchment and during the dry condition (15[th] April - 15[th] October), TTDpk's average is 1.9± 0.9h. Those results are actually in agreements with the lag times to runoff peak of 1.5h mentionned in Iffly et al. 2018. Furthermore, in the updated manuscript, we added TTDpkvalues for the 2016 and 2018 flash flood events, which are 0.9 ± 0.1h and 0.1h respectively. Those values still corresponds to the order of magnitude of the HM section's TTDs during the 15[th] April - 15[th] October period

The study is weakened by the assumption of a constant runoff coefficient through the duration of rainfall (mentioned in line 200 and elsewhere). This seems particularly inappropriate for long events of several days duration, such as were examined in this ms., and even for events of a few hours

duration, when breaks in rainfall (e.g. shown in Figure 6 and Figure 7) allow soil drainage and the re-invigoration of soil infiltrability. It would have been interesting and informative to have seen at least some preliminary sensitivity testing to see how important an effect a changing runoff coefficient might have been to the hydrologic modelling. Perhaps the authors have done such tests and could comment?

We haven't made sensitive analysis before getting your comment. Without making it exhaustively, we tested on the KOE catchment – i.e the one that seems to be more affected by the constant RC assumption – a variable RC along the event. Two Runoff Coefficients – $RC^m$ and $RC^p$ – have been defined: the first one characterizes the re-invigoration of the soil drainage at the beginning of the events and the second one characterizes the hydrological response in the heart of the flood respectively. Arbitrarily, the re-invigoration period is fixed to 20 h. $RC^m$ and $RC^p$ are calculated as indicated below:

- $$\frac{RC^m}{RC^p} = SCW\,20$$

- $$RC^m = \frac{RC \cdot V^{tot}}{V^m + SCW\,20 \cdot V^p}$$

where $V^{tot}$ is the total rainfall amount, $V^m$ the rainfall amount occurring during the first 20 hours, and $V^p = V^{tot} - V^m$. The impact on the FDC scores are presented in the supplementary materials, table S3. The TTD properties resulting in variable RC are compared to the TTD properties with constant RC on figure S4.

According to the FDC score, there is indeed an improvement of the results with a mean decrease of 2%. More specifically the results are significantly better with a variable RC for 11 out of the 40 events. Those events occurs during the November-May period.

Considering the TTD properties, TTD50 and TTDpk decreases in average by 0.5 h and 0.4 h respectively. The decrease is homogeneous on the data set. A largest difference appears during the April-Mai period, resulting in smaller range of transfer lag times uncertainties. Nevertheless, the seasonal variations of the TTD properties can be similarly observable on both unit hydrograph model simulations. The comments about TTDs properties thus based on the simulations with the constant RC hypothesis are still valid.

We recognize that the simulation results could be improved taking into account variable RC, in terms of scores and absolute values. Nevertheless, we assume – according to the presented test, that the general TTD properties variability observed over the seasons (and which is the subject of the paper) is consistent.

It would also strengthen the argument of the paper if the authors could present some data on hydrophobicity in the forested areas, that they appeal to as a mechanism to account for more runoff there. Was hydrophobicity actually present, or was this not investigated? If present, does it dissipate in longer events, so that perhaps it differentially affects runoff behaviour in short convective events in summer?

Unfortunately we have not carried out any hydrophobicity measurements on site, neither during the period nor elsewhere. We can only answer your question indirectly. There is indeed a notable difference between convective summer events and winter events: for the first one, the maximum intensities arrive at the beginning of the rainfall event, whereas for the latter there is a progressive increase in the intensity of the precipitation over time. We can think that the arrival of strong intensity without an initial humidification on a dry and therefore hydrophobic soil, inhibits infiltration. Runoff is then favoured all the more if the ground is sloping (hydrophobicity prevents water from attaching to the ground AND gravity leads to runoff). However, we have not yet been able to verify this hypothesis. While looking for references on this subject, I read your three interesting papers (Dunkerley, 2012, 2016, 2021), which tend, with experimental justification, to the opposite conclusions (a rain peak at the end of an event favours runoff). Nevertheless, as you

mentioned in your article, your experiments are carried out on a flat terrain, and the results ultimately assess the variable infiltration capacity of soils. In a sloping configuration, there will be a first barrier to this infiltration which is the "adhesion" of the water to the soil surface before infiltration. If there is no adhesion (the hydrophobic property of the soil), then the relief might play a key role. An argue in this hypothesis is the fact that the very fast runoff are only observed on the HM section where steep slopes close to the drainage network are present. Finally, as those statement can not be verified with our current data set, we clearly specify at the beginning of our discussion that this is a plausible explanation that must be subsequently checked:

*"Since our dataset appears to be (too) limited for validating our hypothesis, we propose here a list of plausible explanations – based on examples from scientific literature – for the drastic decrease in response times observed in summer on the HM section, as opposed to the KOE section."* (lines 535 – 538)

The authors identify LAI as an important factor in the hydrologic response (TTD) (lines 491-492). Though without comment, the authors appear to use LAI data from 2002-2006, many years prior to their field data collection. This warrants some comment. Further, the LAI seems to be very small, to judge from Figure 9 (left panel), seemingly the only data presented on this variable. The authors only appear to link LAI to the speculation about litter layers and wettability, evidence for which is not provided. Could the authors offer a fuller comment on why LAI might relate to TTD? Do they consider this to be a real, physical effect, or merely a chance statistical correlation (for instance, via some other seasonally-varying parameter)? Their comments and thoughts would be helpful. They could also perhaps consider presenting LAI data for their catchments (as a map) if they have it available. It would appear to be very variable among fields, forests, etc.

We agree that the use of LAI as an indicator of any influence of the vegetation cycle was awkward. We then removed it and replace it by the calendar day (DAY) as an indicator of the seasonal state of the catchment. Having no specific indicator related to forest litter condition, its role discussed in the discussion is presented as a hypothesis to be tested, as said before.

Finally, I wondered whether there is a role for roofs, roads, drains, culverts, etc., in the catchment response. I do not know this area, but Figure 2 suggests that, at least locally, the villages may have impervious areas that are efficiently drained. The main stream channels also warrant at least some description. Have they been modified, perhaps to flow between artificial banks or walls? How significant is the channel travel time from the upper to the lower catchment? In the same way, landuse could helpfully be described, especially whether fields are tilled seasonally.

Relating to the soil sealing in connection with the presence of urban areas (essentially in Larochette), we know from our field knowledge that rapid flows come from both sides of the lateral tributaries, which are not very urbanised apart from a few villages with a few houses on the plateaus.

The transit time from the top to the bottom of the catchment area would require chemical or isotopic tracing measurements which were not carried out for this study.

Concerning land use, an additional figure has been added in the supplemental materials. The steep slopes are mainly covered by forests, the downstream part of the KOE catchment is mainly grassland. There are only crops (mainly corn) on the marly plateaus of the HM section. The seasonal development could have an influence. Nevertheless the shortest times are obtained at the end of September - beginning of October when corn is most developed. This is why we have not detailed their impact.

Overall, this is a solid study, containing some interesting results. However, I am not sure to what extent these actually bear on the factors accounting for flash flooding.

We added some information about the 2016 and 2018 events which must help to link our result to the flash flood issue. Furthermore we really believe that our study help to highlight how fast and concentrated runoff can be processed in specific catchment, beyond the rainfall properties. Consequently we think to give insights to identify catchment prone to flash flood, even if they are ungauged.

_

Minor errors:
- line 13: should be 30 km 2 (space is required between numerals and symbol for unit of measurement)
- line 46 and throughout the paper: 'et al' should be 'et al.' (as a contraction of et alia)
- line 93: should end sentence with a question mark
- line 120: omit the parentheses
- line 145: Captions are reversed (left to right)
- line 162: should say 'Figure 2 left', not right
- line 180: it would be preferable to refer to time-aggregated data as rainfall rates (they are equivalent mean rainfall rates, not true intensities)
- line 253: again, space required following numerical quantity
- Figure 9: there are two dashed lines, only one is listed in the legend
- line 500: Hortonian (capital H after the family name of Robert Horton)

The minors comments has been applied. Thanks for those detailed corrections.

---

## Referee Report (RR1)

**Audrey Douinot et al. (2022) Flood patterns in a catchment with mixed bedrock geology: causes for flashy runoff contributions during storm events**

The manuscript was significantly sharpened and improved. Still, the authors make strong assumptions, that I only partly share regarding extreme events, e.g. in line 127-128: "We indeed assume that the hydrological reactivity of the catchment is detectable independently of the magnitude of the precipitation." As explained in the first review, catchments can show threshold behaviour at high rainfall magnitude that is not visible at ordinary events. However, since the assumptions are clearly marked as such, they can serve as a starting point for discussions in other publications.

Specific comments:

Line 27: "catchment" instead of "cacthment"

Line 28: "but they diverge" instead of "but diverge"

Line 29: What do you mean with "opposite variations"?

Line 30: What do you mean with "concentrated (+- 48% +-87%)?" I do not understand the numbers

Line 32: Is the water transfer time the same as the TTD? If so, please avoid different names for the same parameter and replace water transfer time by TTD.

Line 73: I would delete "(en)", "(fr)" – this information can be obtained from the reference list.

Line 81: The validity of the sentence "in these catchments [Central Europe] climate forcing is not primarily controlled by topography" depends on the definition of Central Europe. I would say, that the Alps belong to central Europe. There, however, climate forcing is controlled by topography.

Line 91: Do "9 years" have to be expanded as you expanded the data used in this study?

Line 107: "extreme" instead of "extrem"

Line 205: "four" instead of "4"

Line 209: "21.7" instead of "21,7"

Line 215: "Germany) -" instead of "Germany),"

Line 215: "consists of" instead of "consists in"

Line 215: Please do not only report on the rainfall sum, but also on the rainfall duration so that one can infer the rainfall intensity.

Line 217: The fact that the discharge volume for the event of 13 July 2021 is uncertain despite the measured discharge height is probably due to uncertainties in the water level-discharge curve. A short comment on this would be good.

Line 245: "corresponds" instead of "correspond"

Figure 5: Please point out in the figure caption that the y-axes are scaled differently.

Table 2: "[l . km-2 . s-1]" instead of "[L . km-2 . s-1]"

---

## Author Response (AR2)

Audrey Douinot et al. (2022) Flood patterns in a catchment with mixed bedrock geology: causes for flashy runoff contributions during storm events

**General comments:**

The manuscript was significantly sharpened and improved. Still, the authors make strong assumptions, that I only partly share regarding extreme events, e.g. in line 127-128: "We indeed assume that the hydrological reactivity of the catchment is detectable independently of the magnitude of the precipitation." As explained in the first review, catchments can show threshold behaviour at high rainfall magnitude that is not visible at ordinary events. However, since the assumptions are clearly marked as such, they can serve as a starting point for discussions in other publications.

We totally agree that a catchment will react faster when rainfall intensities are high enough. We would like to emphasize that some catchments can be more responsive than others for a given rainfall event. And we indeed conjecture that this higher responsiveness of a specific catchment can already be detected at the scale of smaller events. We propose to detail our conjecture by slightly rewording the following sentence (line 124):

*"We indeed conjecture that the hydrological responsiveness of a specific catchment is detectable independently of the magnitude (i.e., volume and/or intensity) of the precipitation event."*
instead of:*"We assume that the hydrological reactivity of  catchment is detectable independently of the magnitude of the precipitation. "*

**Specific comments:**

Line 27: "catchment" instead of "cacthment" Ok

Line 28: "but they diverge" instead of "but diverge" Ok

Line 29: What do you mean with "opposite variations"? During the May-October period the average response time progressively increases in KOE in contrast to HM.  This has been added to the abstract (line 29):

*"During this period, the average response time increases progressively in the KOE catchment, as opposed to the HM catchment."*

Line 30: What do you mean with "concentrated (+- 48% +-87%)?" I do not understand the numbers. These numbers relate to the peak lag time differences between both periods (winter and summer periods, table 5). This has now been detailed in lines 30-32:
*"The HM catchment exhibits similar TTDs during the mid-October to mid-April period, but they diverge markedly duringthe remaining part of the year, with opposite variations. During the mid-April to mid-October period, the average response time increases progressively in the KOE catchment. This behaviour is in stark contrast to the HM catchment, where response times are significantly shorter (peak discharge delay time decreases by -70% ± 28%) and more concentrated (runoff volume occurring in one hour increases by +48% ± 87%) during the mid-April to mid-October, in comparison to the extended winter period."*

Line 32: Is the water transfer time the same as the TTD? If so, please avoid different names for the same parameter and replace water transfer time by TTD. Ok, „water transfer time" has been replaced by „TTD".

Line 73: I would delete "(en)", "(fr)" – this information can be obtained from the reference list. ok

Line 81: The validity of the sentence "in these catchments [Central Europe] climate forcing is not primarily controlled by topography" depends on the definition of Central Europe. I would say, that the Alps belong to central Europe. There, however, climate forcing is controlled by topography.
That's true, by Central Europe, we were referring Belgium, North-Eastern France, Germany and Poland (except the South), which are only part of Central Europe. We propose to change the denomination to „North Central Europe".

Line 91: Do "9 years" have to be expanded as you expanded the data used in this study?
No, because here we refer to an already published study (Pfister et al, 2017)

Line 107: "extreme" instead of "extrem" ok

Line 205: "four" instead of "4" ok

Line 209: "21.7" instead of "21,7" ok

Line 215: "Germany) -" instead of "Germany)," ok

Line 215: "consists of" instead of "consists in" ok

Line 215: Please do not only report on the rainfall sum, but also on the rainfall duration so that one can infer the rainfall intensity. The duration has been added.

Line 217: The fact that the discharge volume for the event of 13 July 2021 is uncertain despite the measured discharge height is probably due to uncertainties in the water level-discharge curve. A short comment on this would be good. The uncertainty on the high discharge values stems from the fact that the water level has risen to the girder of the bridge. The evaluation of the peak flow has mainly an impact on the evaluation of the runoff coefficient, but less on the transfer time distribution. This is why we considered this as being not necessary to detail further.

Line 245: "corresponds" instead of "correspond" ok

Figure 5: Please point out in the figure caption that the y-axes are scaled differently. ok

Table 2: "[l . km-2 . s-1]" instead of "[L . km-2 . s-1] ok
* * *
**REFEREE 3**
* * *
**General comments**

The authors addressed the reviewers' comments carefully. Although most of my questions were solved, I still doubt if the results in summer small events is applicable to summer large events. Large flash flood such as the events of 22-07-2016 and 31-05-2018 has not been observed in summer from 2019 to 2021 except for 13-07-2021 (Table 2). The event of 13-07-2021 had different characteristics of TTD (relatively high value of Vol1h) from summer small events at the KOE catchment (Figure 9; L396-397). Thus, I thought that rapid response may occur even in the KOE catchment during large summer events although the rapid response occurs only in the HM section during small summer events. This is related to my previous comment that not only TTDs but also runoff coefficients (or peak magnitude) should be considered to provide conclusion for flood risk management.

We thank the reviewer for clarifying his point of view on the strong hypothesis of our manuscript, namely that the hydrological reactivity of catchments revealed from moderate precipitation events will reflect the vulnerability of these catchments to more extreme events. Since this hypothesis was discussed several times by the reviewers, we decided to add a section to the discussion outlining the limitations of this conjecture. We support its validity on the one hand, by the correlation analysis carried out on the KOE catchment, showing no dependence of TTD properties on rainfall characteristics. On the other hand, the events cited (23-06-2020, 13-07-2021) corresponding to the most intense events in terms of intensity do indeed show shorter reaction times by the KOE catchment, but without reaching the reactivity of the HM catchment (which remains twice as fast in its reaction). These events thus show that the vulnerability of the HM basin remains higher, even in the case of these more intense summer events.

We have added the following section to the manuscript to strengthen this point of discussion:

**„5.3 Are the conjectured hydrological processes on the basis of moderate events analysis transferable to extreme events ?**

*An important hypothesis of our study is that the high responsiveness of a catchment can be detected from moderate rainfall events, which means that there is no threshold effect between the intensity of precipitation and the reactivity of a watershed. From the moderate events that we have analysed, we have evidence that the HM basin is more likely to generate fast floods because its hydrological response is about twice as fast and more concentrated in summer (in comparison to the KOE catchment). A first element supporting our hypothesis is the fact that the correlation analysis shows very little dependence of the hydrological responsiveness of the KOE catchment on general precipitation properties. This shows its resilience to precipitation characteristics, and thus a constancy in terms of responsiveness. Note that this statement only holds within the range of variation of the analysed precipitation properties and a possible threshold effect beyond this range of variation cannot be completely excluded. However, three events in the database include precipitation intensities of more than 15 mm in one hour. Without being extreme, this is close to the properties of the flash flood events reported in Table 2 and Figure 4.*

*A closer analysis of the hydrological responses of two of these three events with high intensities shows that the response times of the KOE catchment are relatively shorter than those observed in the summer period: the median TTDpk are 3.75h and 7.25h and the VOL1H are 11.6% and 11.1% for the events of June 23, 2020 and July 13, 2021,*

*respectively (in comparison to an average of 8.5h and 6.7% over the mid-April - mid-October period). It is thus possible that the correlation analysis via Kendall's indices may miss this dependence, by the fact that the strong main dependence on seasonality hides a minor dependence. Furthermore, these events are at the margin of those studied and Kendall's coefficients tend to minimise the influences of specific individual events. Assuming then that this influence is possible (despite the fact that it does not appear within the correlation analysis), we can still compare these response times and concentration rates to those observed for the HM catchment: median TTDpk are 1.30h and 2.0h and VOL1H is 13.9% and 11.9% for the events of 23 June 2020 and 13 July 2021, respectively. Thus, the response of HM remains both more concentrated, and above all more than twice as fast. The HM catchment still appears to be more prone to rapid/flash floods than the KOE catchment. Finally, if we compare the response times observed during flash floods in 2016 and 2018 (TTDpk = 0.1h and 0.9h; TTD50 = 5.2h and 1.1h, respectively), they are equivalent to the lowest response times observed in the HM catchment (TTDpk = 0.5h and TTD50 = 2h), supporting the fact that the high rainfall intensities of flash floods did not unequivocally generate faster runoff than moderate events (although the magnitude is not mentioned). Based on the correlation analysis and on the most intense summer events of our data set, we tend to conclude that the high responsiveness of the HM catchment (in comparison to the KOE catchment) prevails during more intense rainfall events and therefore corroborates our initial hypothesis."*

**Specific comments**

L375-376: Why did an event with TTD50 and TTDpk of >20h exist in May (Figure 9) although the authors wrote 17.7h and 13.7h for the maximum value of TTD50 and TTDpk, respectively? You are totally right. To my remember, we used the 5th and 95th percentile to give the range of variation, but the extrema are more apropriate in the quoted sentence. We therefore changed for the extrema values.

Figure 9: Vol1h of flash flood event in July is not shown in the figure. If you could observe the value, please add it to the figure. Also, it would be helpful if it is written in the figure caption that the flash floods were observed at Larochette.
VOL1h of the flash flood of 22.07.2016 is extremely high (64.3 %) and makes the variation of the others not visible, if included in the graph. As the figure is mentioned in table 5, we decided to maintain the display scale of the graph. Nevertheless, in order to be more explicit

on this extreme flash flood value, we added an arrow which points out this score out of the panel. This has been also mentioned in the figure legend:

„*Figure 9: Properties of the simulated transfer time distributions: the median transfer time (TTD50 [h], left panel), the peak flow lag time (TTDpk [h], center), the runoff response concentration in one hour (VOL1h [%], right panel). The events are ordered by calendar day. The orange and green envelopes correspond to the average calendar values, based on the 3 closest estimates and taking into account the uncertainties of the metric (TTD50, TTDpk or VOL1h) assessments.* The purple arrow on the third panel points to VOL1H of the flash flood of July 2016 which exceeds the graph scale (VOL1H = 64.3%).*"*

Figure 10: Does soil moisture mean volumetric soil water content? If so, 90% seems too high. Did you conduct calibration of Campbell CS650 using soil in the study site?

Having four soil water time series with different saturation levels in winter (Figure below, left), we indeed calibrated the time series, according to the porosity and the field capacity of each soil, to get the soil saturation level [%] (Figure below, right). Figure 10 indeed shows the soil saturation level and not the soil moisture content. This has been reworded in the legend:

„Figure 10. Left: The catchment state at the start of each event (points): The minimum discharge during the 7 days before the event (Qbase, [$m^3.km^{-2}.s^{-1}$]), and the soil saturation at 20cm in depth (SWC20 [%]). The light blue colour corresponds to the weekly average discharge minimum at Koedange (solid line) and Medernach (dashed line) over the studied period. The red line corresponds to the soil saturation calendar day average at 20 cm depth in the Medernach catchment over the same period."

[Figure]

Left: raw soil water content measurements at the four raingauge stations (from north to south); Right: standardized soil moisture preserving the temporal variation and fixing the maximum value to 100 %.

**Technical corrections**

Figure 4: "May" instead of "mai". This has been corrected.

Table 5: I think expression like "16 Oct – 14 Apr" is more understandable than "16.10-14.04". ok. This has been reworded.
* * *
**#REFEREE 1**
* * *
This paper has been somewhat improved (and the title modified) over the previous version which was the basis of my report in February 2022. It is well-written and clearly argued, and I enjoyed reading it and considering the arguments raised.

Nevertheless, whilst it includes some interesting and informative material, the paper is somewhat frustrating to read. The authors spend a considerable portion of the paper describing the runoff modelling approach and the rain, soil moisture and discharge data.

We agree that a large part of the manuscript is dedicated to explain the methodology and the data, while the interest of the study is the analysis and the comparison of the TTD distribution given by the model, i.e., from the section 4.2. Nevertheless, we feel that this long section is necessary, both to understand the hypotheses and to

validate the results before using them. The various reviewers' comments also supported this view, as additional descriptions were added in response to questions from the reviewers.

But then it emerges that none of this really sheds much light on the flash flooding whose understanding was the stated objective of the work. As I suggested in my previous review of the ms., what is needed is analysis of the conditions resulting in the flash flooding events (e.g. rainfall intensities, durations, convective storm cell movement, etc.); the focus on far less intense and longer-lasting 'moderate' events seems inappropriate, at least to some extent, and unhelpful. As I pointed out previously, the very high rainfall intensities that seem to have been involved in the flash flooding might account for quite different runoff mechanisms and source areas. For instance, on those occasions, where was the most intense part of the convective rainfall located within the catchments?

The stated aim of our study is to find out „*what is influencing the specific flash flood event patterns, beyond the extreme rainfall properties*" (lines 105 – 106). We assume that it is not only the intensity of rainfall but also the properties of the catchments that will induce rapid runoff to the outlet. Furthermore, we assume that this high reactivity is in fact already present during moderate events, and thus we can base our study on average events to detect the reactivity of a watershed. This assumption of the article is presented in line 126: "*We indeed conjecture that the hydrological responsiveness of a specific catchment is detectable independently of the magnitude (i.e., volume and/or intensity) of the precipitation event* ". It is true that considering this assumption is contrary to the usual assumption of a threshold effect on precipitation intensities. We have therefore added a section in the discussion questioning this strong assumption: '*5.3 Are the conjectured hydrological processes on the basis of moderate events analysis transferable to extreme events ?*' In particular, we detail the fact that the reactivity on KOE appears to be independent of precipitation characteristics, and that on HM the shortest response times (TTDpk = 0.5h and TTD50 = 2h) are close to those observed during the flash floods of 2016 and 2018 (TTDpk = 0.1h and 0.9h; TTD50 = 5.2h and 1.1h), thus demonstrating that the threshold effect on the reactivity is not clearly present on this case.

The authors adopt a number of simplifying assumptions, such as constant event runoff ratio. But given their focus on catchment physiography and geology, I wonder whether instead of studying the hydrologic response of whole sub-catchments, sub-division of contributing slopes into sandstone, marl, etc., might not be necessary to understand the hydrologic response. In this context, the paper lacks

a discussion of the possible spatial variation of soil moisture and hence of runoff response. The authors employ only four soil moisture monitoring sites in the Ernz Blanche catchment (> 100 km2) or just one observation point for ~ 25 km2, and one appears to be located right on the catchment divide, rather than within the catchment. Might there not be zones of preferred runoff production along lower slopes, for instance, towards which gravity drainage might focus seepage and maintain higher soil moisture levels? In other words, are the available data sufficient to properly explore the hydrologic responses of the sub-catchments, or of hillslopes?

The locations of the soil moisture stations were chosen to be spatially representative and on different soil types, but not along a topographic profile. The measurements show few differences during the events suggesting similar infiltration processes. However, the 4 soil moisture measurement stations are located on the plateaus, and indeed the data are insufficient to explore the hydrological response at the slope scale. We therefore mention it at the beginning of the section 5.3 (5.4 in the new manuscript):

*„Since our dataset appears to be (too) limited for validating our hypothesis, we propose here a list of plausible explanations – based on examples from scientific literature – for the drastic decrease in response times observed in summer on the HM section, as opposed to the KOE section."*

This leads the authors to speculate about what happens during dry summer conditions: possible soil hydrophobicity, the occurrence of biomat flow, etc. But they have no data on any of this, only noting in passing toward the very end of the paper (line 604) that some saturation overland flow had actually been observed. I looked for much more discussion about what the authors might have observed in the catchment during their fieldwork: for instance, did they see evidence of overland flow at many places?

It should be noted that the catchment area is located 1 hour's drive from the laboratory and was therefore little visited at the time of the rainfall. Nevertheless, runoff was observed twice during the event of 12 June and during the event of 26 June 2020. In the first case, runoff was observed upstream of Hessemillen on the slopes (between 10% and 20%) in the meadows bordering the main river. In the second case, runoff was observed on the urbanised plateau of the right bank at the Heffingen station. On the other hand, we got conductivity measurements of the water stream at the Medernach station and at the Koedange station on 12th August. The conductivity time series tend to show a dilution in the first cited station and a concentration in the other one case, thus suggesting different contribution processes (surface for dilution and subsurface / underground for concentration). These observations are not part of an exhaustive search (the KOE

basin for example has not been covered during rain episodes), so this is difficult to discuss them in the manuscript.

Do baseflow recession curves shed any useful light on the groundwater storage volumes, nature of the flow, and so on?
We have not thought of applying a baseflow recession curve analysis because we are primarily interested in the flow transfer time. Baseflow recession curve analysis will help to determine the proportion of base flows, but does not contribute to the scope of the study. Furthermore, it appears difficult to apply such an analysis on HM area, as it is a catchment section.

Other comments or queries in my original report seem not to have been addressed at all: the possible role of built drainage systems, pipes, and drainage, and the nature of the channels (have they been aligned, smoothed, etc., for management purposes?).
We apologize for not to having taken into account all your comments. The catchment area is little urbanized as a whole, but the major part concerns the HM section, and in particular its riparian zone (Figure FS2), which can play an important role. Therefore, we added the following sentences lines 591 – 595:
"*. In support of the role of the riparian zone as a buffer to rapid flow, we notice the same hydrograph patterns (Figure 5) at the following hydrometric station (Heffingen, Figure 1), suggesting similar hydrological processes, whereas a river restoration project has been carried out along the hydraulic section between Koedange and Heffingen to improve the lateral connectivity with the major river bed. In contrast, the riparian zone of the HM section is narrower and more urbanized, which further limits the presence of buffer zones for surface runoff, such as wetlands.*"

The possible role of moving convective rain cells is also not mentioned.
We do not mention the possible impact of the movement of the rain cell in the direction of flow of the catchment area, because we want to explain here a systematic difference in speed and not by event.

Despite the limited light shed on flash flood processes in the present paper, in lines 621-623 the authors first suggest that further effort needs to be devoted to simulation tools, focussing on catchment physiography and landscape characteristics:
"When targeting an improvement in flash flood  forecasting in Luxembourg, our results suggest that the focus should be set on the development of a simulation tool adapted to catchments with physiographic characteristics similar to those of the HM sub-catchment – i.e., with fractured bedrock and limited riparian zones."

I found this to sit oddly with their finding that in fact simulation tools - as used in the present paper - proved to shed little if any light on the actual presence of hydrophobicity, biomat flow, etc., which the authors conjecture might account for (or at least make an important contribution to) the flash flooding. Nevertheless, they do go on to suggest that processes of this kind also require investigation.

Perhaps it was a misunderstanding, in that we wanted to say that the improvement of flash flood forecasts - which are always carried out from models, must specifically focus on catchments with similar characteristics to that of HM. As this is mainly related to the forecasting target, we deleted "understanding" adjective.

In the end, it was not clear to me to what extent sub-catchment physiography and geology, as detailed in this paper, are actually important contributing factors to flash flooding (as distinct, for instance, from exceptional rainfall intensities) or might rather exert a lesser, more subdued influence on catchment response than being critical determinants of it. Nevertheless, the paper is potentially a useful vehicle for highlighting some issues that remain unclear, even if it leaves the main hypotheses (lines 109-116) essentially unresolved as far as they relate to the conditions required for the occurrence of flash flooding.

Although we have few examples of flash floods in our database, we can compare the response times of the two catchments with those obtained for flash floods (table below). It can be seen that the impact of the catchment area of occurrence (between resilient = KOE, or vulnerable = HM, Larochette) is more important than the intensity of the events. For an accurate comparison, the impact of extreme events on KOE should also be known. This comparison is now detailed in the section 5.3, and we hope support the relevance of our study.

Table : Range of TTDpk (h) depending of the storm events intensities and the catchment

|  | Resilient catchment (KOE) | Catchment prone to fast runoff (HM, Larochette) | Difference between resilient and vulnerable catchment |
| --- | --- | --- | --- |
| Moderate rainfall events during 15 Apr. – 15 Oct. | 8.5 [3.7 – 12.0] | 1.9 [0.5 – 4.6] | 6.4 [2.4 – 10.7] |
| Flash flood events (2016, 2018) | ? | 0.5 [0.1 – 0.9] |  |
| Difference between moderate and flash flood events | ? | 1.4 [-0.4 – 4.1] |  |

There are a few minor errors:
- line 26: Mars should be March
- line 68: Miller et Dunne should be Miller and Dunne
- line 104: extrem should be extreme
- line 155: supplementary should be supplementary
- line 196 40rainfall-ruoff should be 40 rainfall-runoff
- line 200: use decimal point or comma (not both)
- line 409: propertiesof should be properties of

Thank you for the spelling corrections. We made the corrections as suggested.

line 495 (and many places in the ms.) I think that 'monotonous' should be 'monotonic' This has been reworded.

Figure 10: right panel horizontal axis needs units (mm). A reference to the section detailing the different properties has been added in the legend.

Figure 10: What are the coloured zones in the right-hand panel (not mentioned in the caption of this Figure)? We do not understand which coloured zones you are referring to. If it is the blue-gray background, it highlights the significant correlation (strong correlations (3 stars) as well as weak ones (one star)). But this is already mentioned in the legend.

---

## Author Response (AR3)

Dear editor,

Thank you for accepting my last version of my manuscript.

I uploaded all the material needed to be published in their last version.

Note that I only change the figure labelling (they were two Figure 10) since the last publication. I maintain the table 4 in the manuscript as it is commented on at length in the text (The table S4 is not the same).

Best regards,

Audrey Douinot